# i-shaped antibody engineering enables conformational tuning of biotherapeutic receptor agonists

Matthew G. Romei [1,7], Brandon Leonard[1,7], Zachary B. Katz [2], Daniel Le[3], Yanli Yang[1], Eric S. Day[4], Christopher W. Koo[5], Preeti Sharma [1], Jack Bevers III [1], Ingrid Kim[1], Huiguang Dai[1], Farzam Farahi[1], May Lin[6], Andrey S. Shaw [2], Gerald Nakamura[1], Jonathan T. Sockolosky [1] ✉ & Greg A. Lazar [1] ✉

The ability to leverage antibodies to agonize disease relevant biological pathways has tremendous potential for clinical investigation. Yet while antibodies have been successful as antagonists, immune mediators, and targeting agents, they are not readily effective at recapitulating the biology of natural ligands. Among the important determinants of antibody agonist activity is the geometry of target receptor engagement. Here, we describe an engineering approach inspired by a naturally occurring Fab-Fab homotypic interaction that constrains IgG in a unique i-shaped conformation. i-shaped antibody (iAb) engineering enables potent intrinsic agonism of five tumor necrosis factor receptor superfamily (TNFRSF) targets. When applied to bispecific antibodies against the heterodimeric IL-2 receptor pair, constrained bispecific IgG formats recapitulate IL-2 agonist activity. iAb engineering provides a tool to tune agonist antibody function and this work provides a framework for the development of intrinsic antibody agonists with the potential for generalization across broad receptor classes.

Therapeutic activation of target receptors can be an enormously impactful pharmacologic mechanism for the treatment of disease. Natural ligand-based protein drugs that activate the erythropoietin, growth hormone, insulin, incretin, interferon, and interleukin pathways illustrate the therapeutic benefit of cell surface receptor agonism[1–7]. Correspondingly, the clinical success of these specific examples is a consequence, in part, of the developability of the biological ligands themselves as drug products. However, there are a plethora of cell surface receptors with therapeutic potential as drug targets for which their natural ligands are good research reagents but have poor drug-like properties that limit therapeutic utility. Possible

hurdles include weak protein stability and/or solubility, complex glycosylation patterns, and unfavorable pharmacokinetics (PK) and/or distribution. Furthermore, engineering approaches to improve the developability of endogenous ligands may be offset by the risk of immunogenicity and consequent risk of cross-reactivity against the endogenous protein.

Monoclonal antibodies are the most clinically successful class of biotherapeutics and generally do not suffer from the same limitations as other protein-based drugs. Despite their macromolecular complexity, antibodies typically possess favorable stability and solution properties, limited and well-defined carbohydrate modifications,

[1]Department of Antibody Engineering, Genentech Inc., South San Francisco, CA, USA. [2]Department of Research Biology, Genentech Inc., South San Francisco, CA, USA. [3]Department of Microchemistry, Proteomic, Lipidomics, and Next Generation Sequencing, Genentech Inc., South San Francisco, CA, USA. [4]Department of Pharma Technical Development, Genentech Inc., South San Francisco, CA, USA. [5]Department of Structural Biology, Genentech Inc., South San Francisco, CA, USA. [6]Department of Protein Chemistry, Genentech Inc., South San Francisco, CA, USA. [7]These authors contributed equally: Matthew G. Romei, Brandon Leonard. ✉e-mail: jsockolo@gmail.com; gregorylazar@gmail.com

favorable PK, and relatively low immunogenicity with little evidence of endogenous cross-reactivity. Moreover, decades of drug development experience have resulted in extensive research capabilities for discovery and optimization, and process capabilities for downstream production, purification, formulation, and delivery. Mechanistically, antibodies are clinically validated as competitive inhibitors, mediators of immune effector function, delivery of toxic agents, and more recently immune redirection[8].

In contrast, antibodies have not been as broadly successful as receptor agonists that mimic endogenous ligand activity. A principal challenge for this class of drugs is that the mechanisms by which natural ligands activate receptors are diverse and sometimes insufficiently understood to enable first principal design of active agonists. For example, the ligands of most TNFRSF members induce receptor homo-trimerization when expressed in soluble form and higher order clustering when tethered to a membrane[9]. That stated, agonism activity has been observed for bivalent molecules[10–13]. Conversely, most cytokine receptors require heterodimerization of two receptors in-cis in order to elicit functional signaling[14]. Ultimately, receptor geometry (i.e., the distance and orientation of receptors in relation to one another) can be a key determinant of signal transduction. In the context of antibody agonists, factors that can influence geometry include the bound epitope on the receptor, as well as the orientation, proximity, and rigidity of fragment antigen binding (Fab) regions with respect to each other.

Here, we leverage previously described intramolecular Fab-Fab homotypic interfaces[15,16] to develop a powerful engineering platform to adjust the geometry by which IgG Fab arms engage target receptors. The described approach converts the conventional Y-shaped macromolecular structure of an antibody into a more compact i-shape in which the two Fab arms of an IgG associate to access a unique constrained Fab conformation. We demonstrate the broad utility of these constrained i-shaped antibody (iAb) formats in both monospecific and bispecific contexts by applying them to antibodies against members of the tumor necrosis factor receptor superfamily (TNFRSF) and IL-2 cytokine receptors, respectively.

## Results

### Structural determinants of previously described i-shaped antibodies (iAbs)

Recent studies have characterized a subset of broadly neutralizing HIV antibodies (bnAbs) isolated from infected humans and rhesus macaques that share a unique linear i-shaped conformation that is distinct from the conventional Y-shape[15,16]. These antibodies have a decreased paratope-paratope distance driven by intramolecular association between Fab domains. Physiologically, the Fab-Fab homotypic interaction simultaneously increases the avidity of the interaction with viral surface glycans and generates additional paratopes at the Fab-Fab interface that are essential for antiviral activity.

Thorough characterization of these i-shaped antibodies (iAbs) revealed distinct, independently evolved mechanisms that facilitate iAb formation. One human antibody isolated from an HIV patient, referred to as 2G12, achieves an iAb conformation through heavy chain variable (VH) domain exchange between Fabs (Fig. 1a, left)[15,17]. In SHIV-infected rhesus macaques, two antibody lineages were identified, referred to as DH851 and DH898, with an affinity-driven intramolecular Fab-Fab homotypic interaction between VH domain β-strands A, B, D, and E (Fig. 1a, middle and right)[16]. Both of these mechanisms involve non-covalent Fab-Fab association mediated through distinct yet topologically similar inter-VH interfaces.

Previous mutational work on the 2G12 antibody revealed specific residues contributing to domain exchange[15,18,19]. Consistent with structural studies, the implicated residues are located exclusively in the VH domain and have little to no impact on antigen binding. With these results as a guide, we selected a set of VH residues from 2G12,

referred to as $iAb_{dx}$, that we hypothesized were most critical for inducing the domain-exchanged iAb conformation (Fig. 1b). Similarly, we designed a set of residues based on structural analyses and previous studies[16] predicted to mediate the Fab-Fab affinity interaction in the DH851 and DH898 lineages and termed them $iAb_{aff1}$ and $iAb_{aff2}$, respectively (Fig. 1b). Several of the amino acids found at these positions are not germline encoded and are significantly underrepresented with regard to known human antibody sequences (Fig. 1b), suggesting that they arose via somatic mutation.

### Engineered residue grafts induce iAb conformation

Because antibodies share a high degree of sequence and structural homology, we hypothesized that mutating the sites identified above within a given antibody of interest would be sufficient to induce a similarly constrained iAb conformation to the bnAbs above. To test our hypothesis, we grafted each putative iAb-inducing residue set ($iAb_{dx}$, $iAb_{aff1}$, and $iAb_{aff2}$) into a panel of 10 distinct anti-OX40 antibody clones with diverse sequences, germline precursors, epitopes, and affinities (Fig. S1)[20]. OX40 is a therapeutically relevant TNFRSF member, for which receptor clustering is known to play a role in activation[9]. With the exception of a single CDRH2 mutation in $iAb_{dx}$, the CDRs of all 3 engineered versions ($iAb_{dx}$, $iAb_{aff1}$, and $iAb_{aff2}$) are identical to the parental Abs, and thus have the same target specificity. For comparison, we also produced all 10 anti-OX40 antibodies as contorsbodies, which is a recently described conformationally constrained format that uses genetic linkers to fuse the heavy and light chains of the Fabs to the N- and C-termini of the Fc domain, respectively[21].

The impact of the engrafted iAb residues on antibody conformation was assessed by negative stain electron microscopy. 2D classes from representative anti-OX40 antibody clones with each residue set were compared to those of wildtype (WT) IgG and a representative contorsbody clone (Fig. 2a). Each iAb yielded 2D classes clearly showing two Fabs interacting in parallel and resembling previously reported 2D classes of 2G12 and the macaque broadly neutralizing SHIV antibodies[15,16,22]. The images of the $iAb_{dx}$ clone contained a distribution of conformations with 29% of the particles as i-shaped antibodies and the remaining 71% as standard Y-shaped antibodies. For both the $iAb_{aff1}$ and $iAb_{aff2}$ residue sets, approximately 64% of the particles adopted the iAb conformation, while the standard Y-shaped IgG conformation was observed for the remaining particles. All antibodies with the $iAb_{aff1}$ residue set were monomeric, which was confirmed with analytical size exclusion chromatography (SEC) (Fig. S2). We observed the presence of an iAb dimer in the $iAb_{aff2}$ images where Fabs are associated in an intermolecular head-to-head manner (Fig. 2a, far right image). This result is consistent with analytical SEC results (Figs. S2, S3A, and S3B). Concentration-dependent studies of monomer:dimer ratios support the noncovalent nature of these affinity-based $iAb_{aff2}$ dimer interfaces and suggest the $iAb_{aff2}$ dimer is in dynamic equilibrium (Fig. S3C). As expected, none of the wildtype IgG images were found to have the unique iAb shape, and the contorsbody adopted the previously reported barrel-like conformation with the Fabs pinned next to the Fc (Fig. 2a)[21].

### iAb reformatting enables intrinsic OX40 agonist activity

Like many TNFRSF members, standard antibodies against OX40 generally do not intrinsically promote signaling, but rather rely on extrinsic crosslinking to drive receptor clustering via Fc engagement with cell surface Fc gamma receptors (FcγRs), crosslinking using secondary antibodies, or coating on beads or plates. However, recent studies have increasingly highlighted the importance of rigid molecular formats for TNFRSF agonists[10,23–26]. To determine whether iAb formation could enhance the activity of the engrafted anti-OX40 antibodies, we tested their activity in a cell-based assay utilizing a Jurkat cell line engineered to express OX40 and a nuclear factor κB (NF-κB) luciferase reporter (Jurkat$^{OX40\text{-}NF\text{-}\kappa B\text{-}Luc}$; Fig. S4a). The

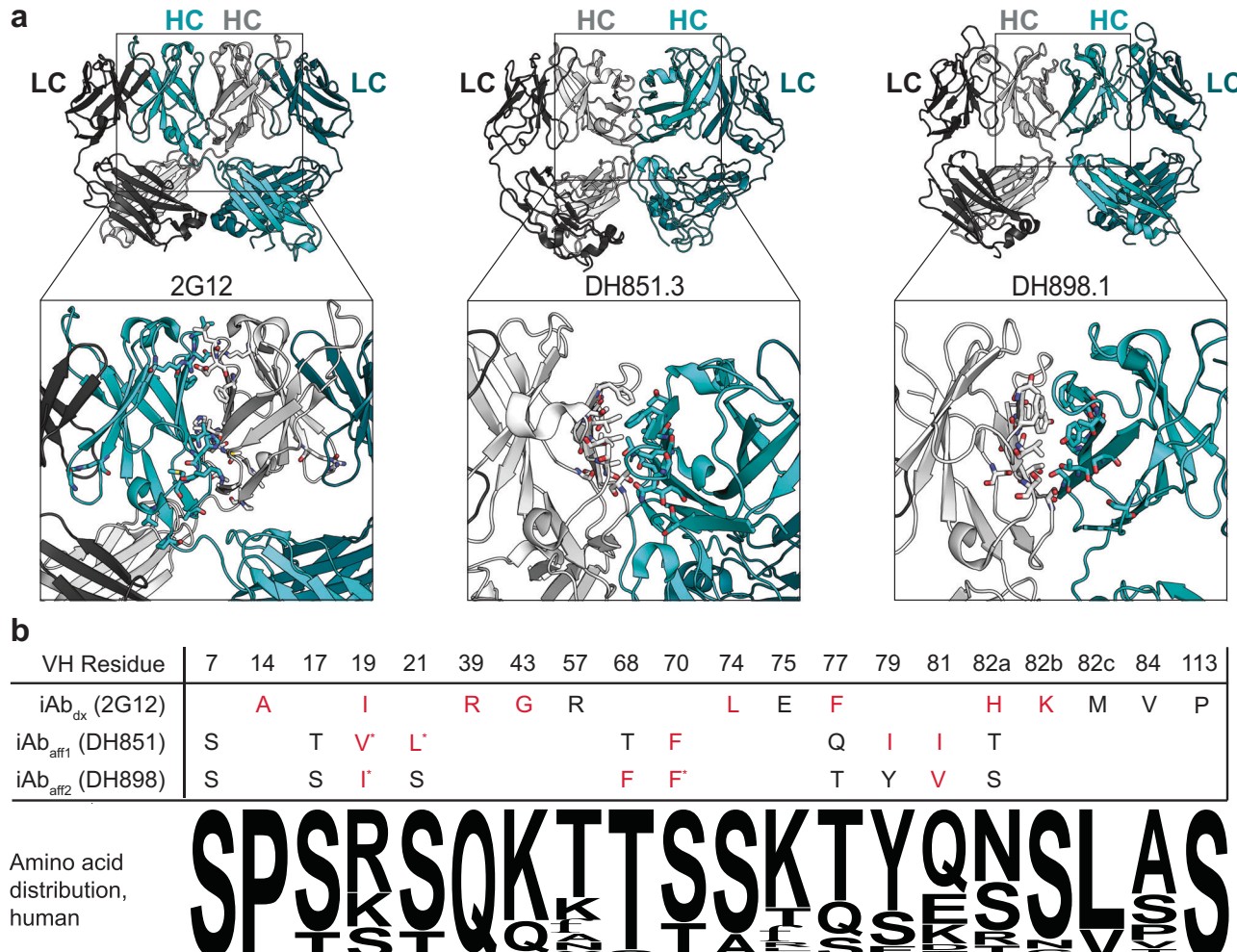

| VH Residue | 7 | 14 | 17 | 19 | 21 | 39 | 43 | 57 | 68 | 70 | 74 | 75 | 77 | 79 | 81 | 82a | 82b | 82c | 84 | 113 |
|---|---|---|---|---|---|---|---|---|---|---|---|---|---|---|---|---|---|---|---|---|
| iAb$_{dx}$ (2G12) | | A | | I | | R | G | R | | | L | E | F | | | H | K | M | V | P |
| iAb$_{aff1}$ (DH851) | S | | T | V* | L* | | | | T | F | | | Q | I | I | T | | | | |
| iAb$_{aff2}$ (DH898) | S | | S | I* | S | | | | F | F* | | | T | Y | V | S | | | | |

Amino acid distribution, human

**Fig. 1 | Structural and sequence analysis of i-shaped antibodies (iAbs).**
**a** Structural representation of the domain-exchanged iAb 2G12 (left, PDB: 2OQJ) and two representative iAbs from the DH851 and DH898 affinity interface lineages, DH851.3 (middle, PDB: 7LU9) and DH898.1 (right, PDB: 7L6M). The light chain (LC) and heavy chain (HC) from each Fab is labeled within each structure. Insets highlight the interface between the heavy chain variable (VH) domains of the two Fabs with residues involved shown as sticks. **b** Table of amino acid residues in the VH domain predicted to contribute to the iAb conformation. VH domain residues are shown using Kabat numbering. The sequence logos below the table are based on the distribution of amino acids at the indicated residue across all human antibody sequences within the abYsis database. Residues shown in red are rare mutations present in <1% of all deposited human sequences. Asterisks denote additional, non-native hydrophobic substitutions previously shown to strengthen the affinity interfaces from the DH851 and DH898 lineages. A representative example of iAb mutation grafting onto anti-OX40 antibodies is shown in Fig. S1.

corresponding WT IgGs and contorsbodies were tested as comparators. Antibody clones expressed as monospecific WT IgG or contorsbody formats had little to no activity on Jurkat$^{OX40-NF-κB-Luc}$ cells (Fig. 2b). Conversely, OX40 agonism activity was observed across all iAb formats with the affinity interfaces, iAb$_{aff1}$ and iAb$_{aff2}$, showing the strongest and most consistent gain of function (Fig. 2b and S5). Only 4/9 iAb$_{dx}$ engrafted clones had intrinsic agonist activity (clone 2A3 did not express), while all iAb$_{aff1}$ and iAb$_{aff2}$ engrafted clones demonstrated improvement in activity over WT IgG controls comparable to that of the native ligand (Fig. 2c). Given the variable activity of the iAb$_{dx}$ panel (Fig. 2b and S5A) and the dimer contaminant observed in iAb$_{aff2}$ samples (Fig. 2a, S2, and S3), we focused on the iAb$_{aff1}$ graft for the remainder of the study.

To determine if the enhanced activity of the engineered iAbs was due to impacts in antigen binding, we performed surface plasmon resonance (SPR) to measure solution-phase monovalent binding affinity. WT IgG and iAb$_{aff1}$ engrafted clones had both similar affinities (K$_D$) and normalized R$_{max}$ (nR$_{max}$) values across the entire anti-OX40 panel tested, indicating that the iAb$_{aff1}$ residue set does not affect affinity or the ability of both antibody Fabs to bind

OX40 simultaneously (Fig. 2d and S5C). In addition to this solution-based analysis, we also performed FACS-based cell binding experiments to determine whether avidity was affected by the iAb$_{aff1}$ conformation upon antigen recognition on the cell surface. WT IgG and iAb$_{aff1}$ samples for each anti-OX40 clone had similar cell binding in terms of both EC$_{50}$ and maximum signal (E$_{max}$) (Fig. 2e and S6). Collectively these data indicate that the intrinsic agonism activity of the iAb$_{aff1}$ format is not a consequence of altered affinity or cell surface avidity.

## The iAb interface is capable of driving OX40 agonism as an intermolecular interaction

To further characterize the iAb interface, we were interested in determining the strength of the homotypic Fab-Fab interface. Because binding measurement is confounded by the intramolecular nature of the interaction in the context of a bivalent IgG, we expressed the iAb$_{aff1}$ engrafted 3C8 anti-OX40 clone as a monomeric Fab and performed sedimentation velocity analytical ultracentrifugation (SV-AUC) to estimate the solution affinity of the homotypic iAb$_{aff1}$ interface. Fitting of the data to a model for monomer-dimer equilibrium confirmed that

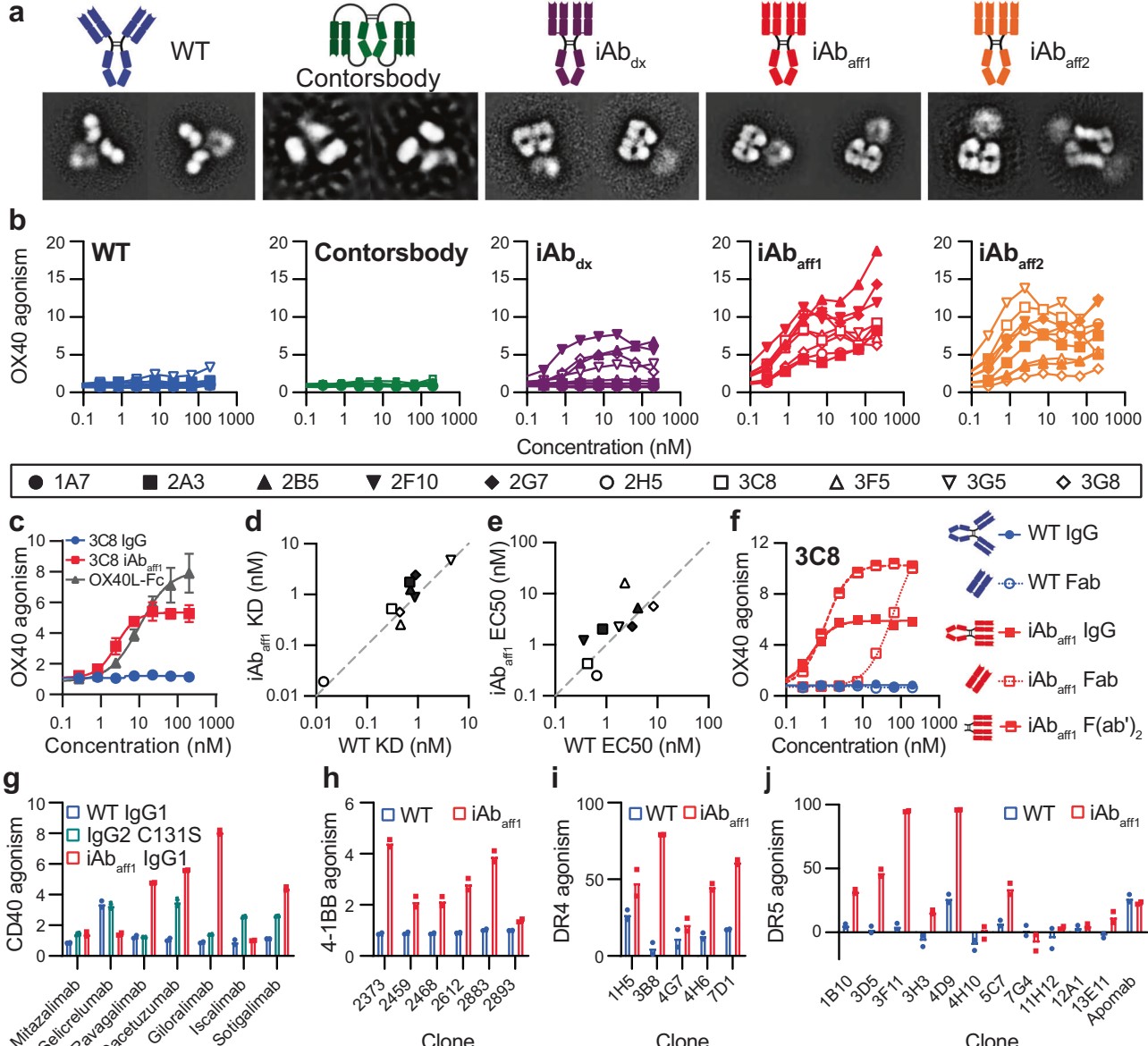

**Fig. 2 | Characterization of iAb induced TNFRSF agonism. a** Cartoons and corresponding representative negative stain electron microscopy 2D classification images for each antibody format. All of the formats in the images contain the variable region of the 3C8 anti-OX40 antibody as representative. **b** OX40 agonism activity from a panel of ten anti-OX40 antibodies for the indicated antibody format. Each symbol represents a unique clone. Data is shown as fold change over an untreated control ($n = 2$ independent wells). **c** Comparison of OX40 agonism activity by the anti-OX40 iAb_aff1 clone, 3C8, to that of the native ligand of OX40, OX40L. Data is shown as fold change over an untreated control ($n = 3$ independent wells) and presented as mean values±SEM. **d** Surface plasmon resonance (SPR) affinity data comparing $K_D$ values for each anti-OX40 clone as either an iAb_aff1 or WT IgG. The dotted gray line has a slope of 1 and indicates no change between the two formats. **e** Cell surface binding to OX40+ Jurkat cells for each anti-OX40 clone

comparing the $EC_{50}$ values of the iAb_aff1 and WT IgG formats. The dotted gray line has a slope of 1 and indicates no change between the two formats. **f** Effect of antibody fragmentation on OX40 agonism activity with and without iAb_aff1 mutation set engraftment for a single anti-OX40 clone, 3C8 ($n = 2$ independent wells). **g–j** TNFRSF agonism activity of various formats of clones against CD40 (**g**, 4 nM), 4-1BB (**h**, 22.2 nM), DR4 (**i**, 100 nM), and DR5 (**j**, 100 nM). Data are shown at a single concentration (indicated above in paretheses) taken from titration curves in Fig. S9 ($n = 2$ independent wells). CD40 and 4-1BB agonism are shown as fold change over an untreated control, while DR4 and DR5 agonism are shown as % killing relative to an untreated control. Each CD40 clone was produced as a WT IgG1, IgG2 C131S, and iAb_aff1, while all clones against other targets were only produced as a WT IgG1 and iAb_aff1. Source data are provided as a Source Data file.

the 3C8 iAb_aff1 Fab forms a reversible dimer in solution with a $K_D$ of approximately 6.8 µM (Fig. S7 and Table S1).

Since the 3C8 iAb_aff1 Fab was capable of forming a dimer in solution, we asked whether the iAb_aff1 interface could enable agonist activity in a monovalent format through intermolecular interaction (Fig. 2f). Indeed, the iAb_aff1 interface still enabled intrinsic agonist activity as a monomeric Fab, albeit at an $EC_{50}$ nearly 100-fold weaker than the corresponding full-length iAb_aff1. The greater than an order of magnitude discrepancy between the $EC_{50}$ of the 3C8 iAb_aff1 Fab in this

cell-based activity assay and the affinity determined by SV-AUC can likely be explained by the increased local concentration at the cell surface driven by receptor binding[27]. Strikingly, the maximal activity was increased almost 2-fold for the iAb_aff1 Fab relative to iAb_aff1 IgG. Intrigued by this result, we produced and tested a F(ab')₂ version of the 3C8 iAb_aff1 antibody and observed a similarly heightened level of maximal activity that had an $EC_{50}$ equivalent to iAb_aff1 IgG (Fig. 2f). These results suggest that the presence of the Fc region in a full-length IgG may sterically restrict iAb_aff1 activity. Curiously, we observed no

i-shaped conformation for the 3C8 iAb$_{affl}$ F(ab')$_2$ upon analysis with negative stain electron microscopy (Fig. S8) despite the same covalent linkage of the Fabs through IgG1 hinge disulfides. Though IgG versions of the iAbs were our primary focus moving forward for their more favorable drug development properties (discussed further below), the iAb engineered (Fab')$_2$ and Fab versions contribute insights into the relationship between macromolecular interactions and activity. While further investigation is needed to understand mechanism, these results indicate that iAbs can enable intrinsic agonism through both intra and intermolecular Fab interaction, as discussed more extensively in the Discussion.

## iAbs broadly enable intrinsic TNFRSF agonism

After demonstrating the ability of the iAb conformation to intrinsically agonize OX40, we were eager to explore whether the same constrained conformation could be generalized to enable agonism against other TNFRSF members. We produced 4 panels of publicly available and in-house derived antibodies with diverse sequences, germline precursors, and affinities against CD40, 4-1BB, DR4, and DR5 as both WT IgG and iAb$_{affl}$ formats. Because previous studies of anti-CD40 antibodies have shown enhanced agonist activity as human IgG2 isotypes with disulfide rearrangements that restrict hinge flexibility[23,28,29], we also produced the C131S variant of IgG2 that promotes the constrained h2B isoform as a comparator for CD40.

For a majority of the antibody clones across all targets, iAb$_{affl}$ reformatting enhanced agonist activity compared to the corresponding WT IgG format, which generally showed little to no agonism (Figs. 2g–j and Fig. S9). For CD40, there was no apparent correlation between a given antibody's activity in the different formats (WT IgG, IgG2 C131S, iAb$_{affl}$) and its previously reported epitope[28–31]. However, the strongest agonist activity was observed for iAb$_{affl}$ formats of ravagalimab, dacetuzumab, giloralimab, and sotigolimab (Fig. 2g). The most consistent results were observed for 4-1BB agonism, where 5/6 iAb$_{affl}$ clones had intrinsic agonist activity at least 2-fold greater than the untreated control, in contrast to the corresponding WT IgGs that were all inactive without crosslinking (Fig. 2h). For both CD40 and 4-1BB, iAb reformatting resulted in equivalent or enhanced activity compared to the native ligand of each receptor (Fig. S10). DR4 and DR5 agonism proved to be more variable with only 3/5 anti-DR4 iAb$_{affl}$ clones showing a 2-fold increase in activity over WT IgG controls and 5/12 anti-DR5 iAb$_{affl}$ clones able to kill at least 25% of cells (Fig. 2i, J). Overall, these results highlight the broad applicability, albeit empirical in nature, of the constrained iAb format as an engineering tool to generate effective intrinsic agonists against TNFRSF members.

## Comparison between bivalent iAbs and highly avid hexameric IgG

Activation of many TNFRSF members has been shown to benefit from higher order receptor clustering mediated by extrinsic crosslinking. Because of this, engineering approaches to increase avidity, such as IgG hexamerization[32], have generated effective antibody-based agonists[24,33,34]. Given the enhanced activity of the iAbs compared to WT IgGs in the absence of a discernable difference in binding properties, we were interested in comparing our conformation-based agonist mechanism of action to that of avidity-based agonism. To do so, we introduced mutations in the Fc-domain of the 3C8 antibody known to induce hexamerization and drive intrinsic OX40 agonism (Fig. 3a)[20,24,32]. In the Jurkat$^{OX40-NF-κB-Luc}$ reporter assay, the 3C8 iAb$_{affl}$ had a similar EC$_{50}$ with slightly lower maximum signal compared to the hexameric IgG format (Fig. 3b), which is striking given that hexameric 3C8 has a valency of 12, while the 3C8 iAb$_{affl}$ only has a valency of 2. In addition to activity, we also assessed the effect of these formats on receptor-mediated internalization by labeling the antibodies with a pH-sensitive dye that increases in fluorescence under low pH conditions such as those present in acidified lysosomes. Using flow

cytometry to detect and quantify fluorescence, we observed that the hexameric IgG drives substantially increased internalization relative to WT IgG, concomitant with its high intrinsic activity (Fig. 3c). Interestingly, while the iAb$_{affl}$ similarly mediates strong intrinsic OX40 agonism, albeit slightly less than hexameric IgG, the associated level of receptor downregulation is modestly greater than the inactive WT IgG. While further study is needed, these results offer the prospect that different engineering approaches to intrinsic agonism can have different activity versus internalization profiles, the consideration of which would have clear importance for biotherapeutic design.

We further explored the mechanistic impacts of OX40 agonism mediated by the 3C8 iAb$_{affl}$ and hexamer using total internal reflection fluorescence (TIRF) microscopy to track clustering patterns and single particle mobility of fluorescently-tagged OX40 following treatment of transiently transfected Jurkat cells with the antibody formats. Max projections for the entire 12.5 s acquisition under each treatment condition show a largely diffuse distribution of OX40 for both the untreated and WT 3C8 treated cells, whereas iAb$_{affl}$ and hexamer treatment induce hotspots of receptor accumulation with the hexamer having a more punctate pattern (Fig. 3d). Inset windows with the molecular trajectories illustrate how each molecule moves and provide additional clarity on individual molecular confinement. Mean square displacement analyses of the trajectories further highlight the differences between the formats and clearly show that the hexamer restricts movement of OX40 compared to the other formats (Fig. 3e). While the iAb$_{affl}$ conformation still allows more free 2D diffusive movement of OX40 similar to untreated cells and WT 3C8 (Fig. 3e), closer analysis of the individual track intensity values show a shift in the distribution for the iAb$_{affl}$ similar to that of the hexamer (Fig. 3f). Taken together, these data indicate a propensity for both the iAb and hexamer to tightly link two or more receptors, explaining the increased activity and internalization driven by these two formats. Yet, there may be mechanistic differences, as the hexamer induces higher order, larger scale clusters of OX40 with greatly reduced mobility within the membrane, potentially explaining the greater activity and more pronounced internalization caused by treatment with this format.

## Discovery and characterization of IL-2Rβ and IL-2Rγ binders as a path to cytokine mimetic antibodies

Given the success of applying iAb engineering to monospecific agonists across multiple TNFRSF targets, we explored whether iAbs could also be applied to bispecific antibody agonists as a strategy to agonize heterodimeric receptors. Many disease-relevant cytokines signal through classical receptor heterodimerization[35]. We focused on the IL-2 pathway given its therapeutic relevance in both cancer and autoimmunity[14,36] and increasing interest in IL-2 mimetics[37–40]. Interestingly, previously reported approaches utilize IL-2 receptor (IL-2R) binding moieties that are smaller than antibody Fab-arms, and, to our knowledge, bispecific antibody agonists of heterodimeric cytokine receptors based on the traditional IgG format have not been reported. Thus, we were interested in exploring the impact of constrained bispecific antibody formats on IL-2 pathway agonism.

IL-2 forms a signaling complex with 3 receptors: IL-2Rα, IL-2Rβ, and IL2Rγ. IL-2Rβ and IL-2Rγ are responsible for downstream signaling upon heterodimerization, while IL-2Rα stabilizes the complex and enhances IL-2 potency[41–43]. For this application, IL-2Rβ and IL2Rγ binders were discovered from an in-house, naïve, fully human scFv library (Fig. 4a)[44]. After multiple rounds of selection each with increasing stringency (Fig. S11), we selected 61 anti-IL-2Rβ and 34 anti-IL-2Rγ scFvs with unique sequence families for reformatting as monospecific IgG and characterization for cell surface receptor binding, affinity by SPR, and ability to block IL-2 signaling (Fig. 4b, Fig. S12 and Table S2). The monovalent binding affinities ranged from 4 - 74 nM and 7 - 131 nM for anti-IL-2Rβ and anti-IL-2Rγ antibodies, respectively. Eight anti-IL-2Rβ and 6 anti-IL-2Rγ antibodies were selected for comprehensive epitope

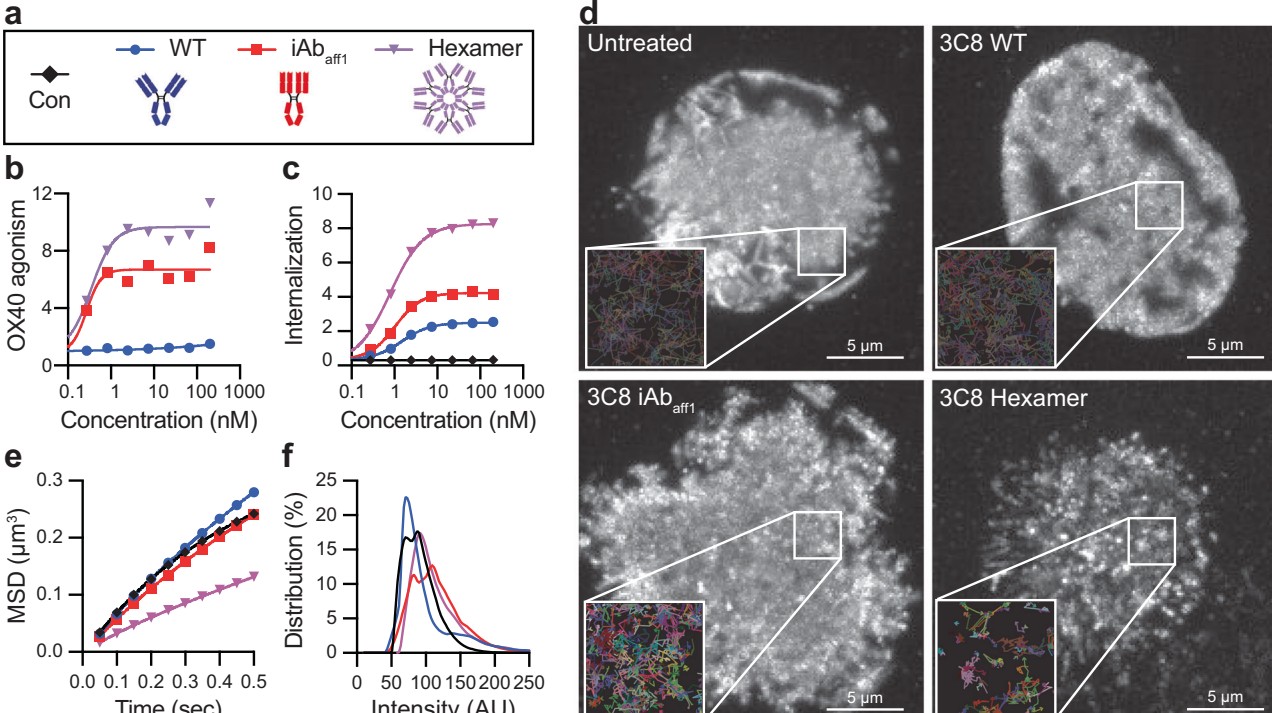

**Fig. 3 | Comparison of bivalent iAb$_{aff1}$ and hexameric IgG. a** Legend with cartoons depicting each format engineered into the anti-OX40 3C8 clone. All plots below (**b**, **c**, **e**, and **f**) are colored according to this legend. **b** OX40 agonism activity of each format ($n = 2$ independent wells). **c** Receptor-mediated internalization of each format shown as MFIx10$^4$ ($n = 2$ independent wells). An antibody against an irrelevant viral antigen (gD) labeled with the same pH sensitive dye is shown as a control in black. **d** TIRF microscopy max projections (grayscale) for a 12.5 second acquisition of a representative Jurkat T cell expressing OX40-mNeonGreen and treated with the indicated antibodies at 13.3 nM in solution. Insets show representative single molecule tracks as arbitrarily colored lines within a $2.5 \times 2.5$ µm area. This experiment was repeated twice with similar results. **e** Average mean square displacement (MSD) plots for all analyzed tracks are shown for each treatment condition. **f** The distribution of the average background subtracted molecular track intensity for each treatment condition. For **e**, **f**, the untreated control is shown in black. Source data are provided as a Source Data file.

mapping based on their ability to bind antigen both via SPR and cell surface binding.

To gain a more detailed perspective on binding epitopes beyond traditional epitope binning by cross-blocking, we utilized a mutational scanning technique in which an alanine was introduced at each residue of the extracellular domains (ECDs) of IL-2Rβ and IL-2Rγ, resulting in 206 and 203 receptor mutants, respectively. Alanine mutations that impacted antibody binding were determined by SPR, mapped onto the crystal structures of IL-2Rβ and IL-2Rγ (PDB ID: 2ERJ)[41], and binding epitopes were inferred by structural visualization of disruptive mutations that clustered in close proximity to one another (Fig. 4c). The 6 anti-IL-2Rγ clones bound dispersed epitopes across the receptor with some commonalities (Fig. 4c, blue box). Clones G02, G25, and G28 overlapped with the IL-2 binding site (Fig. 4c, black box), while G12 and G23 bound near the membrane proximal region of the IL-2Rγ ECD. G33 bound a distinct epitope near the N-terminus of IL-2Rγ. Conversely, all 8 anti-IL-2Rβ clones shared a similar binding region in close proximity to the binding site for IL-2 (Fig. 4c, red box). Clone B30 differed slightly in that its binding site almost completely overlapped with that of IL-2, consistent with the ability of B30 to potently block signaling by IL-2 (Fig. 4c and S12).

**Constrained bispecific antibody formats facilitate IL-2 pathway agonism**

Matrixing of the 8 anti-IL-2Rβ and 6 anti-IL-2Rγ lead clones reformatted as bispecific WT IgG, contorsbodies, and iAb$_{aff1}$ resulted in 48 unique bispecific combinations for each format. The ability of each bispecific antibody to agonize the IL-2R pathway was first quantified using a Jurkat reporter cell line engineered to stably express IL-2Rβ and IL-2Rγ, as well as luciferase under control of a STAT5 promoter (Jurkat$^{βγ-STAT5-Luc}$; Fig. S4C. While we observed no activity for WT IgG combinations, multiple clone combinations were active in both the contorsbody and iAb$_{aff1}$ constrained formats (Fig. 5a). Overall, the contorsbody format had a higher hit rate with 12/48 clones having greater than a 2-fold increase over the untreated control. There was a clonal and epitope dependence on activity, with active anti-IL-2Rγ clones targeting epitopes near the IL-2 binding site (Fig. 4c). It is unclear if this IL-2Rγ region is intrinsically favorable for antibody mimetic activity or if the increased activity of clones binding this region was biased due to the lack of epitope diversity within the anti-IL-2Rβ clones. For the iAb$_{aff1}$ panel, only 3/47 clones were active using the same cutoff (G23/B65 did not express), and all 3 were combinations shared a single IL-2Rβ clone, B10, despite the aforementioned epitope similarity among the IL-2Rβ clones. Interestingly, there was no overlap between active contorsbody and iAb$_{aff1}$ clone combinations. To further explore the activity differences across the panel of iAb$_{aff1}$ bispecifics, we collected images of one active (B10/G28) and one inactive (B09/G28) iAb$_{aff1}$ bispecific using negative stain electron microscopy (Fig. S13). Both bispecific antibodies exist in solution in an equilibrium of i-shaped and Y-shaped conformations (39% i-shaped particles for the former active combination and 59% i-shaped particles for the latter inactive combination). This result indicates that the activity difference is not due to population of the iAb conformation, but rather suggests the importance of other Fab properties such as epitope, binding orientation, and/or binding affinity and kinetics. Taken together, these results underscore the importance of antibody conformation, binding epitope, and receptor geometry on agonist activity.

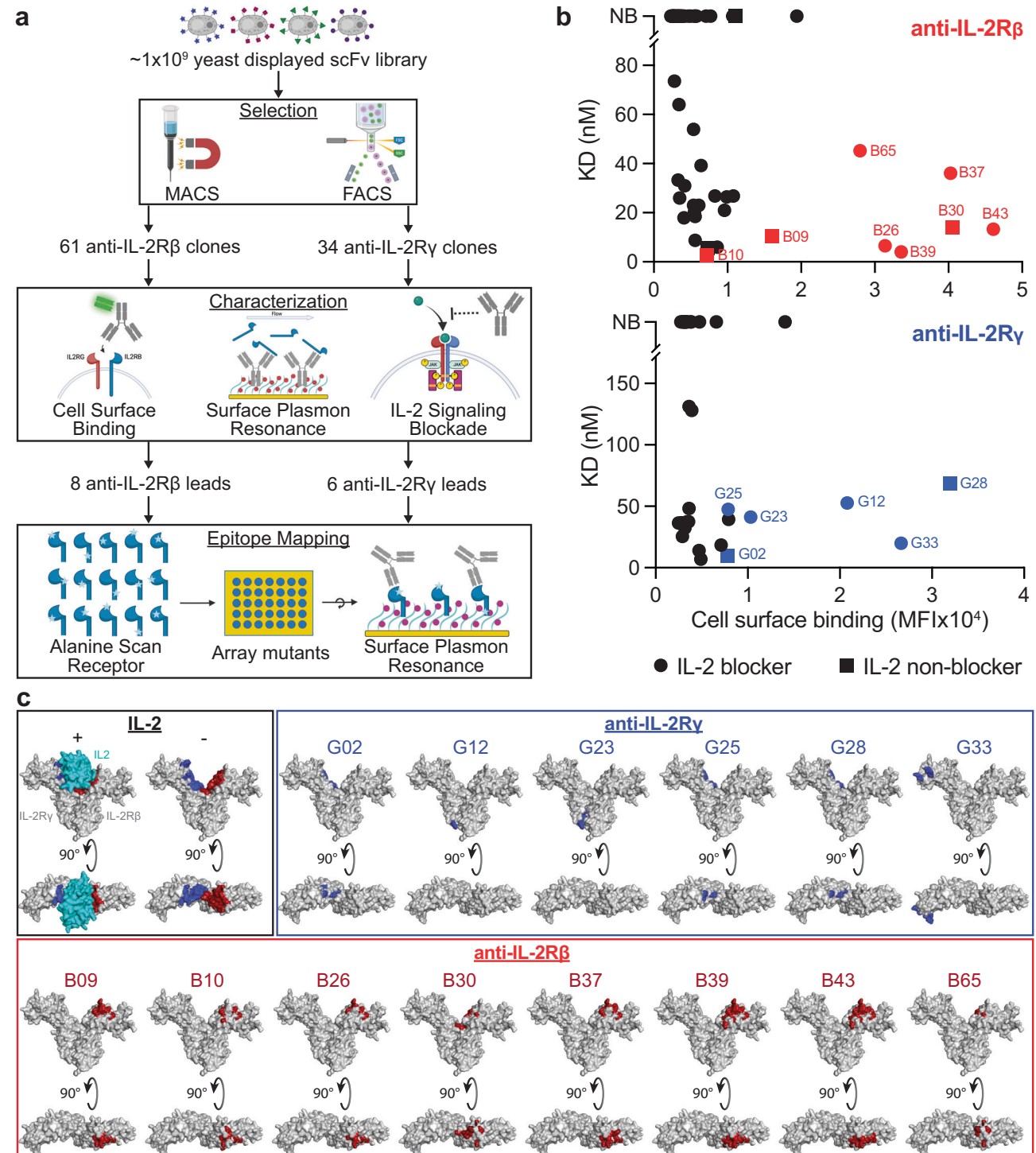

**Fig. 4 | Selection and characterization of anti-IL-2Rβ and anti-IL-2Rγ clones.**
**a** Schematic depicting the selection and screening process for binders to IL-2Rβ and IL-2Rγ (created with BioRender.com). **b** Plots comparing affinities determined by SPR and cell surface binding propensity for 61 anti-IL-2Rβ (top) and 34 anti-IL-2Rγ clones (bottom). Clones that block IL-2 signaling in the Jurkat$^{βγ-STAT5-Luc}$ reporter assay are shown as squares and non-blocking clones are shown as circles. Red and blue symbols indicate selected lead clones for IL-2Rβ and IL-2Rγ, respectively. All other nonlead clones are black. Source data are provided as a Source Data file. **c** Epitope mapping of lead anti-IL-2Rβ (red box) and anti-IL-2Rγ (blue box) clones. Epitopes on IL-2Rβ and IL-2Rγ are shown in red and blue, respectively. For reference, the IL-2 binding site is highlighted based on the ternary complex (black box, PDB: 2ERJ). In each image, the IL-2Rβ and IL-2Rγ structures are shown on the right and left, respectively. A 90° rotated image of the crystal structure is shown to further depict epitope residues.

We selected 2 lead contorsbodies (B09/G02 and B09/G28) and 2 lead iAb$_{aff1}$ (B10/G25 and B10/G28) to characterize further. First, we performed titrations of each lead molecule in the Jurkat$^{βγ-STAT5-Luc}$ reporter assay. Consistent with the single point concentration screen, WT IgG bispecifics failed to agonize the IL-2 pathway, whereas bispecific contorsbodies and iAb$_{aff1}$ exhibited dose-dependent agonist activity with potencies (EC$_{50}$) and maximal activation (E$_{max}$) similar to that of IL-2 (Fig. 5b). To determine whether differential agonist activity of the constrained versus conventional IgG bispecific formats was a result of an increased propensity to simultaneously bind IL-2Rβ and IL-

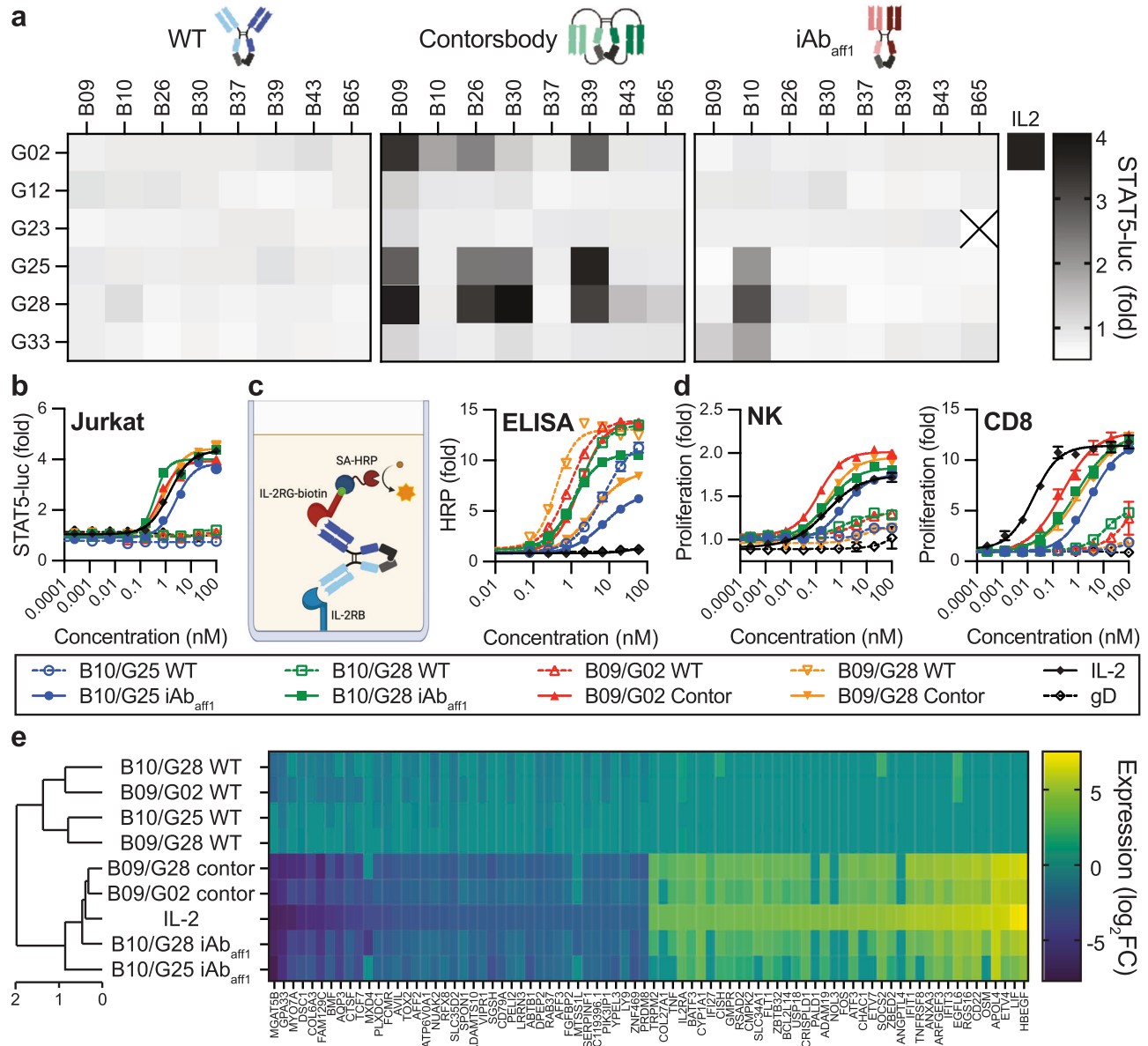

**Fig. 5 | Characterization of constrained IL-2 antibody agonists. a** Heatmaps depicting IL-2R pathway agonism in Jurkat[bg-STAT5-Luc] reporter cells. Jurkat[bg-STAT5-Luc] cells were treated with 100 nM of bispecific WT IgG (left), contorsbodies (middle), and iAbs (right) overnight at 37 °C and IL-2R pathway agonism was quantified by luminescence and plotted as the fold increase in luminescence over untreated controls (n = 2 independent wells). The activity of wild-type human IL-2 at 100 nM is shown for comparison. X denotes no expression. **b** Concentration-dependent activity of lead constrained IL-2 pathway agonists and corresponding WT IgG controls in the Jurkat reporter assay (n = 2 independent wells). **c** Schematic (left, created with BioRender.com) and plot (right) of an IL-2Rg/IL-2Rb bridging ELISA (n = 3 independent wells). The Y-axis reflects fold-change of absorbance signal over control that lacks any antibody sample. **d** Activity of lead constrained IL-2 pathway agonists and corresponding WT IgG controls in primary NK cells (left), and primary CD8 T cells (right) (n = 1 biological replicate/donor per cell type, n = 3 independent wells). Each cell type was obtained from a different individual donor and isolated to >85% purity. For B-D, a legend showing symbols for each line is shown below and an antibody against an irrelevant viral antigen (gD) is shown as a control. For **c**, **d**, data is presented as mean values ± SEM. Source data are provided as a Source Data file. **e** Hierarchical clustering of various IL-2R pathway agonists and corresponding WT IgG bispecifics based on altered gene expression in primary CD8 + T cells determined by mRNA sequencing. Differential gene expression is depicted as the log₂ fold change (log₂FC) of the top 40 up- or down-regulated genes in each treatment group relative to an isotype control antibody (anti-gD) (n = 3 independent samples). Genes are arrayed by column and IL-2 ligand and mimetic antibodies by row.

2Rγ, we performed a bridging enzyme linked immunosorbent assay (ELISA). Constrained formats exhibited a decreased ability to bind to both receptors simultaneously, suggesting that the agonist activity of iAb[aff1] and contorsbodies is not due to increased binding but rather closer receptor proximity (Fig. 5c).

### Constrained antibody agonists mimic IL-2 proliferative activity and transcriptional reprogramming of primary cells

IL-2 is a potent T and NK cell proliferative cytokine[14]. To rigorously interrogate the activity of our top bispecific agonist antibodies and compare to IL-2 cytokine in a more physiologically relevant cell type, we quantified the ability of our lead bispecific combinations to drive expansion of both primary human NK and activated CD8 + T cells in vitro. iAb[aff1] and contorsbody treatment resulted in dose-dependent proliferation of NK cells with agonist activity similar to that of IL-2, whereas the corresponding WT IgG bispecific formats had limited agonist activity (Fig. 5d), consistent with activity on Jurkat[βγ-STAT5-Luc] cells (Fig. 5b). The iAb[aff1] and contorsbody samples also promoted dose-dependent expansion of CD8 + T cells, albeit with reduced potency compared to IL-2 (Fig. 5d). The increased potency of IL-2 compared to

the IL-2 mimetic iAb$_{aff1}$ and contorsbodies is likely due to the expression of IL-2Rα on activated T cells[43], which, as mentioned above, acts as an affinity-enhancer for IL-2.

Finally, we characterized the gene expression profiles induced by IL-2 and bispecific antibody agonists in CD8 + T cells by mRNA sequencing (Fig. 5e). Hierarchical clustering of normalized differentially expressed genes revealed two major groups of treatment conditions, with the lead constrained formats closely associated with IL-2 in one group and the WT IgG controls in a distinct group. A heat map depicting the 40 most up- and down-regulated genes by IL-2 revealed a strong overlap with the constrained formats, but not the WT IgG controls. Generally, similar trends were observed when comparing agonism activity in the Jurkat and primary cell assays in Fig. 5a, b, d with the induced gene expression levels in Fig. 5e. The B10/G28 bispecific iAb promoted stronger agonism and more pronounced changes in gene expression levels than B10/G25; however, neither signatures were as strong as IL-2. Collectively, these data demonstrate that constrained antibody formats enable receptor agonism of otherwise inactive bispecific antibody combinations in a manner that mimics the native ligand at both the proliferative and transcriptional level.

## Discussion

Approaches to discover antibodies with agonistic activity have thus far been largely empirical and often reliant on high-throughput antibody discovery capabilities. Some agonist antibodies discovered through traditional means have been explored in the clinic with mixed results[45]. However, antibody discovery is only a first step, and agonist antibodies can be rare, have only nascent potency, and/or rely on extrinsic factors for activity[46]. There is a need for generalizable engineering approaches to facilitate agonist antibody discovery against disease relevant receptors to accelerate preclinical and clinical development.

Beyond antibody binder discovery, a variety of rational engineering approaches have been explored to improve antibody agonists[46]. Parameters for tuning an IgG for agonist activity include affinity, valency/avidity, and conformation. Affinity is readily adjusted with modern molecular engineering techniques, and previous studies have demonstrated that, at least in select cases, there can be a dependence of receptor activation on antibody affinity[20,47,48]. Avidity can be highly impactful, particularly in the case of targets for which clustering is a critical element of receptor activation[49]. Approaches to enhance avidity include increasing valency by linking paratopic binding units through genetic fusion or conjugation, multivalent formats such as Fc hexamers and IgM, as well as engineering of homotypic antibody interfaces to enhance self-association[10,20,24,33,34,50,51]. More recently, conformation has been explored as an optimizable property to vary the geometric nature of antibody/receptor engagement[52]. Successful agonism generally relies on both the proximity and orientation of paratopes with respect to one another and conformational sampling (i.e., the distribution of structural conformations of the molecule in solution). Some examples include IgG hinge modifications[23,28], polymer-based Fab-Fab linkers[53], tandem single and/or variable domain formats[24,25,38,54], engineered single domain formats[39,55], and complex constrained formats[21]. While these aforementioned strategies have demonstrated success in vitro and, in some cases, pre-clinically, many deviate significantly from the conventional IgG format that has become a proven therapeutic modality, thus adding molecule risk to an already high-risk therapeutic hypothesis with complex pharmacology.

In this work, we describe a robust conformational tuning approach for engineering IgG-like antibody agonists and demonstrate its utility for two unrelated classes of therapeutically relevant receptor targets. iAbs require a small number of VH framework mutations (Fig. 1), minimizing perturbations that, together with its conventional IgG nature, offer a path for designing antibody agonists in a developable format. The mutation sets are derived from naturally occurring

antibodies that bind repeat glycans on the HIV envelope. While it cannot be completely ruled out that the mechanism of action of the engineered agonists in the present study involves glycan binding, the diverse nature of the panels of antibodies and protein targets that we explored suggests otherwise. Both domain exchange and affinity interface mechanisms can lead to iAb formation and enhanced agonist activity, with the iAb$_{aff1}$ residue set modeled after the previously described DH851 lineage offering the most promising combination of biophysical properties and agonist activity in both monospecific and bispecific formats. The success of the iAb format as a monospecific agonist of five TNFRSF members (Fig. 2) and potent bispecific mimetic of the common gamma chain cytokine, IL-2 (Fig. 5), demonstrates its potential for broad applicability.

Native TNFRSF ligands are trimeric and agonize receptors via formation of 3:3 ligand:receptor complexes and/or higher order clustering[56]. Accordingly, much of the work to improve antibody-based agonists for this receptor class has been to enhance their ability to cluster receptors by increasing valency as discussed above. The iAb approach represents a unique method of receptor agonism and reinforces a growing body of evidence suggesting that bivalency can be sufficient and potentially beneficial for agonizing some TNFRSFs[11,13]. Our comparative analysis of the iAb$_{aff1}$ and high valency IgG hexamer formats indicate that these distinct molecular approaches achieve intrinsic TNFRSF agonism with differing effects, possibly through dissimilar mechanisms. Our data show that the IgG hexamer contributes to greater perturbation of OX40 membrane kinetics, including greater receptor internalization, increased subdomain cluster formation, and decreased mobility (Fig. 3). The results further suggest that local receptor clustering as indicated by the higher OX40 track intensity upon treatment with the iAb$_{aff1}$ is sufficient to drive OX40 agonism activity, consistent with prior work demonstrating that small but dense receptor clusters are most important for TNFRSF signaling[48,57].

The above discussion becomes more nuanced when one considers that the observed enhanced activities of the engineered iAbs may be a consequence of a complex interplay of intra- and inter-IgG interactions. The electron microscopy and SEC data indicate that iAb$_{aff2}$ forms inter-molecular dimers, whereas only monomeric species are observed for iAb$_{dx}$ and iAb$_{aff1}$ (Fig. 2a, S2, and S3). Yet the sedimentation data indicate that the monomeric iAb$_{aff1}$ Fab forms weak reversible inter-molecular dimers in solution that associate in the μM range (Fig. S7 and Table S1). The weak potency of that iAb$_{aff1}$ Fab (although still intrinsically active relative to inactive WT Fab) provides a quantitative contribution to potency for the inter-molecular interaction (Fig. 2f). IgG versions of the iAbs, our primary focus for drug development (discussed further below), are of course capable of both intra- and inter- molecular interactions, as evident by iAb$_{aff2}$. While we were able to observe dimer formation of iAb$_{aff2}$ by analytical SEC (Fig. S2 and S3), the reversible nature of the interaction precludes preparative separation of monomer and dimer or higher order species. Thus, for our lead platform iAb$_{aff1}$ IgG, which we only observe as a monomer, it is not possible to directly dissect the contributions to activity of a putative oligomeric population. The iAb$_{aff1}$ IgG is substantially more potent relative to the μM inter-molecular dimer affinity as determined for the iAb$_{aff1}$ Fab (Fig. 2f). In the simplest explanation, the intra-molecular iAb conformation that is evident for iAb$_{aff1}$ IgG1 is the dominant contributor to activity, and thus its intrinsic activity likely reflects access to unique conformational receptor engagement. In an alternate interpretation, activity may be dominated by inter-molecular interactions that are driven to high potency by avidity of IgG/receptor complexes[27,49]. Such a model would be akin to Fc hexamer variants, where single substitutions that are apparent monomers in solution display enhanced activities similar to the triple RGY variant that forms hexamers in solution[34,58]. While these two interpretations are not mutually exclusive, and both intra- and inter-molecular

interactions may contribute to the enhanced activities of iAbs, our TIRF microscopy analysis indicates that there is a clear mechanistic difference between receptor agonism induced by iAbs and avidity-based hexameric IgG.

In addition to the molecular engineering strategies discussed above, epitope can also impact agonist potential[28,54]. Correspondingly, our application of iAbs and contorsbodies to bispecific cytokine mimetics shows a strong dependence on epitope. Comprehensive epitope mapping revealed similar epitopes for all anti-IL-2Rβ clones, while the anti-IL-2Rγ clones had a wider range of epitope diversity (Fig. 4). However, all active IL-2 mimetics, with the exception of one iAb_aff1 clone combination, bound epitopes near the IL-2 binding site on both IL-2Rβ and IL-2Rγ. The distance between the optimal epitope combinations ranged between 3-5 nm, which is consistent with paratope-paratope distances of ~5-8 nm and 4.8 nm for the contorsbody and DH851, respectively[16,21]. Despite the epitope similarity between all anti-IL-2Rβ clones, there were additional clone combinations with similar epitope-epitope distances that were not active agonists as either iAb_aff1 or contorsbodies. These results indicate that epitope is necessary but not sufficient, and suggest that other factors, for example the orientation by which the Fab interacts with receptor, also determine optimal activity.

While this study demonstrates proof-of-concept for the use of the iAb platform for agonist antibody design, unknowns remain. For example, we found that conformational tuning using different engineering approaches had variable success across applications. When comparing iAbs and contorsbodies, the iAb_aff1 format was a potent agonist across all anti-OX40 clones tested, while the contorsbodies were entirely inactive. In contrast, more bispecific clone combinations had IL-2 mimetic activity in the contorsbody than the iAb format, and there was no overlap of favorable epitope combinations. Within target receptor classes there was variability as well. For TNFRSF members, the iAb_aff1 format differed in its ability to agonize OX40, CD40, 4-1BB, DR4, and DR5 both in terms of potency and the number active clones. While precise biochemical understanding requires deeper study, we speculate that these differences are due to the conformational differences between the iAb and contorsbody formats, the potential of the iAb format for both intra- and inter-molecular interaction, and the mechanistic diversity of receptor activation both within and across receptor families.

Considerable investigation also remains from the standpoint of drug development. Any impact of the introduced mutations and iAb conformation on developability or in vivo performance requires examination, including for example effects on solution properties, target specificity/polyreactivity, pharmacokinetics, and immunogenicity. Molecular assessment and mitigation approaches for these important developability elements have been extensively reported[59–64]. A favorable aspect of iAb engineering is that it enables enhancement to an otherwise IgG-like drug format, for which there is compellingly the strongest level of development and clinical validation. Indeed, the established process development and favorable PK of IgG's are the reasons we focused on the full-length IgG version of iAb_aff1, in contrast to a would-be rapidly clearing F(ab')₂ version, despite the latter's apparent higher level of maximum agonist activity for anti-OX40 (albeit equivalent potency to iAb_aff1 IgG). In the broader picture, translation of any new biotherapeutic format approach takes substantial resources, time, and risk. For example, antibody drug conjugates (ADCs) and bispecific antibodies were in research and development for decades before clinical validation as biotherapeutic classes. With the present study, our hope is to enable antibodies to achieve an eventual similar level of success as receptor agonists. Despite clinical advancement of an extensive pipeline of TNFRSF agonist antibodies, including against OX40 and the other targets explored here, as yet none have been approved, and several have been withdrawn. While diverse factors may be responsible, including foremost the biology of individual receptors, the general reliance on extrinsic Fc-mediated crosslinking for activity represents a major deficiency of this class of agents[9]. For agonists of cytokine receptors, the poor developability, rapid clearance, and in some cases pleiotropy of native cytokines have motivated recent efforts to discover mimetics based on heavy chain only or tandem nanobody fusions[37,39]. A principal next step for the work here is to explore the translational potential of iAbs to enable intrinsic agonists of TNFRSF, cytokine, and other classes of receptors in a tried-and-true IgG-like format.

From a biotherapeutic engineering perspective, the current work capitalizes on naturally existing, albeit rare, Fab interactions, and in a manner that relies on the high sequence and structural homology of antibodies for design. The coupling of these elements is part of what makes antibody engineering unique and powerful. Together with our previous work investigating and exploiting homotypic antibody interactions[10,20], the current study illustrates the prodigious value of continued investigation into how antibodies naturally interact, with their target and with themselves, to perform their biological function. The iAb platform presented here leverages clever conformational tricks used by the immune system for pathogen neutralization to create a tool for optimization of an important mechanistic class of biotherapeutics.

## Methods

### Molecular cloning

Antibody clones against each target were produced from various sources. anti-OX40, anti-4-1BB, anti-DR4, and anti-DR5 antibodies were discovered internally via mouse immunization campaigns. Sequences for all anti-CD40 antibodies used in this work were derived from publicly available databases and patent literature. The anti-IL-2Rβ and anti-IL-2Rγ antibodies used in this study were discovered in-house using a synthetic, fully human scFv yeast display library, as described below.

Gene fragments encoding all antibody constructs were synthesized as gBlocks or eBlocks (IDT) and cloned into the pRK mammalian expression vector using Gibson assembly (NEB, cat#E2611L). The pRK vector contains a cytomegalovirus (CMV) enhancer and promotor to control gene expression, an N-terminal secretion signal (MGWSCIILFLVATATGVHS), a C-terminal simian virus 40 (SV40) PolyA sequence, and an ampicillin resistance gene for bacterial selection. Unless otherwise stated, all Fc regions were human IgG1 with the effectorless mutations L234A/L234A/P329G (LALAPG; EU numbering). The contorsbodies were constructed by fusing the heavy chain and light chain Fab domains to the N- and C-termini of the Fc domain, respectively, as previously described[21]. In short, the C-terminus of the heavy chain (C220) was fused to the N-terminus of the Fc domain (D221), and the N-terminus of the light chain was fused to the C-terminus of the Fc domain (K447). Each fusion domain was separated by a flexible $(G_4S)_2$ linker.

To make hexameric 3C8, heavy chain variable regions were cloned into an hIgG1 backbone containing the previously described E345R/E430G/S440Y (RGY) mutations[20,24,32]. As the hexameric IgG exists in equilibrium, no additional purification measures were taken to isolate the desired species. To promote formation of the h2B isoform of IgG2 for the anti-CD40 antibodies, the C131S mutation was used (EU numbering, corresponds to the C127S mutation of White and colleagues)[23]. Fab constructs consisted of the light chain paired with a truncated heavy chain (1-225, EU numbering) and a C-terminal TEV protease-cleavage site, and a Flag tag. For all bispecific antibodies, knob-in-hole mutations were introduced into the Fc-domain to enable heterodimerization and prepared as described below[65].

OX40 ECD (L29-D170) was cloned into the pRK mammalian expression vector with a TEV protease-cleavable N-terminal His tag. The IL-2Rβ ECD (A27-T240) and IL-2Rγ (L23-A262) extracellular domains (ECDs) were cloned into the pRK mammalian expression vector with a C-terminal His tag.

## iAb engineering

In this work, the iAb conformation was induced in antibodies of interest through engraftment of specific sets of mutations (Fig. 1b and S1). The residue set used to induce domain exchange (iAb$_{dx}$) was inspired by previous structural and mutational studies on the 2G12 antibody[15,18], and the specific mutations with a representative example of the grafting approach can be found in Fig. 1b and S1. The affinity interface iAb mechanism utilizes a hydrophobic patch on the surface of the heavy chain variable domain to facilitate intramolecular Fab-Fab association. The residue sets used to generate these Fab-Fab interactions and facilitate iAb formation (iAb$_{aff1}$ and iAb$_{aff2}$) were inspired by lineages of broadly neutralizing anti-SHIV antibodies discovered in SHIV-infected macaques, specifically DH851 and DH898[16]. In order to graft each residue set onto "acceptor" antibody clones, we first aligned each antibody sequence and then substituted the amino acids at the given residues in Fig. 1b with the appropriate iAb inducing residue set. A representative example of two anti-OX40 antibody grafts is depicted in Fig. S1. Based on varying degrees of amino acid conservation at each of the residues, grafting the residue set resulted in between 4 to 8 mutations per antibody, with an average of 7 mutations across all antibodies studied in this work.

## Protein expression and purification

With the exception of anti-DR4 and anti-DR5 antibodies, protein expression was performed by transfection of DNA into in-house HEK293 cells. Anti-DR4 and anti-DR5 antibodies drive apoptosis of HEK293 cells and were therefore produced with in-house CHO cells. For IgG and iAbs, co-transfection of heavy and light chain DNAs was performed. Since contorsbodies contain a genetic fusion of the light chain to the Fc region, only a single plasmid was required for monospecific formats. OX40 ECD was expressed with a baculovirus expression system in Tni insect cells in the presence of 10 mg/L kifunensine.

Following expression, affinity chromatography was performed using MabSelect SuRe resin (Cytiva, cat#17543803) for Fc-containing proteins, CaptureSelect CH1-XL resin (ThermoFisher, cat#194346201 L) for Fabs, and NiNTA agarose resin (Qiagen, cat#30210) for the receptor ECDs. Elution buffers consisted of 50 mM sodium citrate at pH 3.0 and 150 mM NaCl for the MabSelect SuRe and CaptureSelect CH1-XL resins, and 50 mM sodium phosphate at pH 7.4, 200 mM NaCl, and 400 mM imidazole for the NiNTA resin. Size exclusion chromatography was used as the final purification step using a HiLoad 16/600 Superdex 200 column. Protein quality was determined by analytical SEC using a Waters xBridge BEH200A SEC 3.5 um (7.8 × 300 mm) column (Waters, cat#176003596) and by SDS-PAGE. All antibody formats were stored in a buffer consisting of 20 mM histidine acetate and 150 mM NaCl at pH 5.5, while the receptor ECDs were stored in 25 mM tris pH 7.5 and 150 mM NaCl.

Bispecific IgG and iAb production was performed, as previously described[66]. In brief, half antibodies containing either the knob (T366W) or hole (T366S, L368A, Y407V) mutations were first expressed in separate cell cultures and purified as described above. Two half antibodies were assembled into a single bispecific antibody through annealing, reduction, and oxidation steps. Annealing of the 1:1 half antibody mass mixtures was performed at 37°C for 25 min followed by 24°C for 30 min. Reduction of the disulfides was performed with the addition of 2 mM dithiothreitol for 2 h. After a 30 min oxidation step using 5 mM dehydroascorbic acid, the desired heterodimer species was separated from unwanted homodimers using size exclusion chromatography. Due to the genetic fusion of the light chain, bispecific contorsbodies were produced in a single cell culture as described above without any in vitro assembly steps.

## Negative stain electron microscopy

Antibody samples for negative stain EM analysis were exchanged into a buffer consisting of 25 mM tris and 150 mM NaCl, concentrated to 1 mg/ml, and filtered through 0.22 μm membranes (Costar, cat#8160). Samples were then diluted to 0.01 mg/ml, and 4 μl of the diluted sample was immediately deposited on a glow-discharged (Solarus plasma cleaner, Gatan) ultra-thin carbon coated 400-mesh copper grid (Electron Microscopy Sciences). After incubation for 30 s, the remaining liquid was blotted away with filter paper (Whatman, cat# WHA1001090), and the grid was washed 5× with 30 μl of filtered 2% uranyl acetate (Electron Microscopy Sciences). The excess uranyl acetate stain was blotted away with filter paper after 30 s. The grids were imaged on a Talos 200 C equipped with a 4 K Ceta CMOS camera (ThermoFisher) at 73,000× magnification (2 Å per pixel). SerialEM was used for all data collection, and image processing was performed with cisTEM analysis software to generate 2D class averages. Percentages of i- and Y-shaped antibodies for a given sample were calculated using the number of particles in each 2D class.

## Generation of F(ab')$_2$

The F(ab')$_2$ construct of 3C8 iAb$_{aff1}$ was generated through proteolytic cleavage of the lower hinge using a modified matrix metalloproteinase 3 (MMP3) as described previously[67]. In brief, the MMP3 protease was fused to the N-terminus of an in-house affinity matured anti-human Fc antibody based on the rheumatoid factor RF61[68]. Additionally, the MMP3 protease was engineered for more efficient activation through the addition of an enterokinase cleavage site within the pro-domain. Pro-domain cleavage and subsequent activation of the MMP3-antibody fusion construct was achieved by incubating 16 units of enterokinase (NEB, P8070L) for every 25 μg protein at room temperature for 16 hours in a buffer containing 25 mM tris at pH 7.5, 150 mM NaCl, and 10 mM CaCl$_2$. To inactivate the enterokinase, 0.1 mg/ml soybean trypsin inhibitor (Sigma, 17075029) was added to the protein solution. The activated MMP3-antibody fusion was mixed with the 3C8 iAb$_{aff1}$ construct at a 1:10 molar ratio and incubated overnight at 37 °C. MabSelect SuRe resin was used to remove all cleaved Fc, unreacted IgG, and MMP3-antibody fusion protein, then the supernatant was purified with size exclusion chromatography and analyzed via SDS-PAGE.

## Sedimentation velocity analytical ultracentrifugation (SV-AUC)

A concentration series of 3C8 Fab with the iAb$_{aff1}$ residue set graft was analyzed by sedimentation velocity analytical ultracentrifugation (SV-AUC) to demonstrate the presence of dimer in solution and determine the affinity of the homodimer. AUC is a well-established method for quantitative analysis of macromolecule interactions in solution. Sedimentation velocity analytical ultracentrifugation (SV-AUC) experiments were performed in an Optima XL-I analytical ultracentrifuge (Beckman-Coulter, Indianapolis, IN) at 20 °C and 50,000 RPM (201,600 g). 3C8 iAb$_{aff1}$ Fab was prepared at 0.6, 0.2, and 0.07 mg/ml in a buffer containing 25 mM histidine acetate at pH 5.5 and 150 mM NaCl and loaded into the sample sector of 2-sector 3 mm charcoal filled EPON centerpieces (Spin Analytical, Berwick, ME) with the diluent buffer in the reference sector. Samples were equilibrated to 20 °C for 2.5 h before the run was started. Sedimentation was monitored at 280 nm using the UV/vis absorbance system on the centrifuge in continuous mode with a radial step size of 0.003 mm. 150 scans were collected for each sample over approximately 8 h. The distributions of the apparent sedimentation coefficient, $g(s^*)$, taken at the same reduced sedimentation time for each concentration were calculated using SedAnal v6.80[69], and the increase in the weight average sedimentation coefficient with increasing concentration is characteristic of an associating system (Fig. S7A).

Subsequently concentration difference data for each protein concentration were globally fit to a monomer−dimer equilibrium model in SedAnal v6.80 (Fig. S7B)[70]. The molecular weight of the monomer, the sedimentation coefficients of the monomer and the dimer, as well as the $K_D$ for the monomer−dimer equilibrium were

floated as variables in the fit. The molecular weight of the dimer is fixed in the model to be twice the molecular weight of the monomer. The results are collected in Table S1. These results confirm that 3C8 iAb$_{aff1}$ Fab forms reversible dimers in solution with $K_D$ ~ 6.8 μM.

### Affinity measurements

Solution affinity constants for all antibodies were determined on a Biacore 8k+ or T200. Antibodies were diluted to 1 μg/ml in HBS-P+ buffer (Cytiva, cat#BR100671) and captured on a Series S Protein A chip (Cytiva, cat#29127555) according to the manufacturer's protocols. Serial dilutions of the appropriate receptor ECDs (recombinantly produced OX40, CD40, 4-1BB, DR4, DR5, IL-2Rβ, and IL-2Rγ, as described above) were prepared in HBS-P+. The dilutions were injected for 3 min, followed by a 5 min dissociation step. Affinity constants were determined from kinetic fits to the sensograms using the Biacore Evaluation Software.

### Cell binding analysis

A 0.6 μM solution of each anti-IL-2Rβ or anti-IL-2Rγ antibody was incubated overnight at 4 °C with 2.4 μM of Alexa Fluor 488 labeled anti-human IgG goat affiniPure Fab fragment (Jackson, cat#109-547-008). Serial dilutions of each 4:1 molar ratio Fab:antibody mixture were prepared in clear 384-well FACS plates in 20 μL FACS buffer (1x PBS with 1% BSA). 20 μL of FACS buffer containing 80,000 IL-2Rβ and IL-2Rγ overexpressing Jurkat cells was added to each well and incubated for 4 h at 4 °C. Cells were pelleted and washed 2 times, resuspended in 40 μL of FACS buffer, and analyzed on an iQue3 cytometer (Sartorius).

### Receptor-mediated internalization assay

WT IgG, iAb$_{aff1}$, or hexameric versions of the anti-OX40 clone, 3C8, were labeled with pHAb amine reactive dye according to the manufacture's protocols (Promega, cat#G9841). OX40 expressing Jurkat cells were treated with the indicated concentration of each pHAb labeled format for 1 h at 37 °C and 5% $CO_2$ in RPMI media containing 10% FBS and 2 mM L-glutamine (cRPMI). Cells were then washed twice with PBS containing 1% BSA and fluorescence was measured using the BL2 channel of a Sartorius iQue3.

### Total internal reflection fluorescence (TIRF) microscopy and single particle tracking

Jurkat T cells were transfected (Amaxa) with 0.3 μg of an OX40-mNeongreen plasmid in a pRK vector backbone 48–72 h before live cell imaging. 48 well glass bottom plates were coated with 100 μg/mL poly-L-lysine for 30 min at 37 °C, washed, and allowed to dry overnight before addition of anti-CD3ε antibodies (OKT3 at 10 μg/mL) to stimulate T cells and enhance spreading. All imaging was performed on a Nikon TIRF system with a 100× 1.49 NA objective, Hamatsu Orca FusionBT SCMOS camera, and iLas2 laser system for ellipse illumination to flatten the field at an imaging depth of 75–100 nm. After cell adhesion to surfaces the pre-treatment datasets were acquired at 20 Hz for 12.5 to 25 s (250–500 frames). The indicated anti-OX40 antibody formats were then added to the imaging wells at 2 μg/mL (13.3 nM) and the same pre-treated cells plus additional cells were acquired at the same frame-rate for the next 10 min. At least 6 cells per condition were analyzed from two independent experiments resulting in over 20,000 OX40 trajectories per condition. Tracking was performed with a DiaTrack 3.0 MatLab runtime application and mean square displacement plots were generated with custom written Igor track analysis code[71]. In brief, image stacks were background subtracted in DiaTack and processed with a gaussian filter based on a 1.2 pixel half-width half-max value for the point spread function. Particle-identification thresholds were user determined to ensure proper identification versus background after previewing 50-100 frames. Tracks with a max displacement of 3 pixels (390 nm) and minimum lifetime of 3 frames (150 ms) were used to generate MSD curves that

average the square displacement of all molecules over the given time frame. Representative max projection images were created with imageJ and track insets were created with Igor and registered to the representative fields in Adobe Illustrator.

### Yeast display

An in-house derived, *S. cerevisiae* yeast displayed scFv library with a diversity of $1.6 \times 10^9$ unique sequences was used to discover binders against IL-2Rβ and IL-2Rγ using a combination of magnetic-activated cell sorting (MACS) and fluorescence-activated cell sorting (FACS) as depicted in Fig. S11. Briefly, yeast were electroporated with plasmid encoding the scFv library and grown to log phase at 30 °C in SD-CAA media. To ensure sampling of the entire library, the number of yeast cells used for each round of selection was 10-fold over the theoretical or measured library diversity. scFv display was controlled via a galactose-inducible promoter. Yeast were grown at 20 °C in SG-CAA media (containing galactose) for 24-48 hours at a starting $OD_{600}$ of 1.0 to induce scFv display. Before each round of selection, scFv expression on the yeast surface was confirmed using an Alexa Fluor 488-labeled anti-cMyc antibody (1/50 dilution, Cell Signaling, cat# 2279). Binding to IL-2Rγ or IL-2Rβ was determined by the addition of the indicated concentration of in-house derived IL-2Rβ containing a C-terminal Avi tag for site specific biotinylation or commercially sourced IL-2Rγ-biotin (Acro biosystems, cat#ILG-H85E8) and streptavidin (SA) beads (Miltenyi, cat#130-048-101) or Alexa Fluor 647-labeled SA tetramers (ThermoFisher, cat#S21374). For initial rounds of selection using magnetic SA beads, biotinylated antigen (500 nM final concentration) was mixed with 1 mL of SA beads prior to addition of yeast to enhance avidity. Subsequent rounds of selection were conducting with increasing stringency using tetrameric SA-antigen (500 nM SA) followed by decreasing concentrations of monomer. For monomer selections, yeast were first incubated with biotinylated antigen, washed with PBS containing 1% BSA and then stained with a 1/1000 dilution of SA. Each round was checked for enrichment of a binding population by staining yeast with a titration of antigen and analyzing fluorescence using an iQue3 (Sartorius). The results for 37 nM are shown in Fig. S11. DNA was extracted from each the final rounds of selection via ZymoPrep (Zymo Research, cat#D2004), transformed into *E. coli*, and individual colonies were selected for sanger sequencing.

### Epitope mapping

Epitope mapping of anti-IL-2Rβ and anti-IL-2Rγ clones was performed using a Carterra LSA. Alanine substitutions were first introduced at each residue of 6x histidine tagged IL-2Rβ and IL-2Rγ ECDs using PCR based mutagenesis (n = 206 and 203, respectively). If an alanine was already present the residue was mutated to glycine. Cystines were not mutated. All mutants were expressed in 293 cells and purified using NiNTA agarose resin as described above. Purified IL-2Rβ mutants were then arrayed and captured on a NiHC200M sensor chip for 5 min. A bispecific format where only one arm was specific for IL-2Rβ was flowed over the chip for 5 min and buffer was flowed for 5 min to allow for dissociation. The chip was then regenerated with 350 mM EDTA twice for 5 min and prepped with 5 mM NiCl for 5 min. This process was repeated for each of the anti-IL-2Rβ lead clones, and with the IL-2Rγ mutants combined with the anti-IL-2Rγ lead clones. Mutant receptor capture levels were calculated for each mutant at each cycle and response unit measurements were taken at the end of the association phase of each antibody. Ligand levels were plotted against antibody binding to identify alanine mutations that impacted antibody binding and these positions were highlighted on the previously reported crystal structure of the corresponding receptor[41].

### Bridging ELISA

Recombinantly expressed human IL-2Rβ was coated onto a Maxisorp 96-well plate (ThermoFisher, cat#44-2404-21) overnight at 4 °C using a

1 µg/ml solution in PBS. The wells were then blocked with a solution of 0.5% BSA and 2 mM EDTA in PBS for 1 hour at room temperature. Three-fold dilutions of the lead antibodies were prepared in PBS with a top concentration of 60 µg/ml, and 100 µl of the antibody dilutions were added to the wells. The plates were incubated for 1 hour at room temperature and then washed 3 times with PBS. During the antibody incubation, a solution of 10 µg/ml biotinylated human IL-2Rγ (Acro Biosystems, cat#ILG-H85E8) and 100 µg/ml streptavidin-HRP (SouthernBiotech, cat#7100-05) was prepared in PBS and incubated at 37 °C for 30 minutes. The IL-2Rγ and streptavidin-HRP solution was diluted ten-fold, and 100 µl was added to the each washed Maxisorp well. After incubation for 30 minutes at 37 °C, the plate was washed 3 times with PBS. 100 µl of TMB substrate (ThermoFisher, cat#N301) was added to each well, and the absorbance at 650 nm was measured after 10 minutes. The absorbance signal was reported as fold-change over a control well without any added antibody.

## Cell-based reporter assays

OX40, 4-1BB, DR4, and DR5 assays were performed as previously described[20,24]. In short, both OX40 and 4-1BB assays used over-expressing Jurkat-NFκB-luciferase reporter cells (from an in-house source and Promega cat#JA2351, respectively) seeded at 80,000 and 40,000 cells/well, respectively, in 20 µl RPMI containing 1% glutamine and 10% heat inactivated fetal bovine serum in 384-well tissue culture plates (Corning cat#3764). Antibody dilutions in 20 µl of the same RPMI medium were added to each well. Plates were incubated overnight at 37 °C and 5% $CO_2$. The level of receptor agonism was determined by quantification of luciferase expression using 40 µl Bright-Glo reagent per well after a 5 min room temperature incubation (Promega, cat#E2650) and a Perkin-Elmer Envision plate reader.

For DR4 and DR5, Colo-205 cells (ATCC cat#CCL-222), which endogenously over-express both DR4 and DR5, were plated at 100,000 cells/well in a white 96-well plate (Corning cat#3917) in 50 µl RPMI supplemented with 1% glutamine and 10% heat inactivated fetal bovine serum and incubated overnight at 37 °C and 5% $CO_2$. The next day, the cells were treated with 50 µl of the indicated antibody at various dilutions. After incubation for 24 hrs at 37 °C and 5% $CO_2$, cell death was quantified by CellTiter-Glo 2.0 (Promega, cat#G9242) using a Perkin-Elmer Envision plate reader.

For the CD40 bioassay, reporter cells were purchased from Promega (cat#JA2151) and used to assess CD40 agonist activity as follows. Cells were thawed and 10,000 cells were plated in each well of a black walled 384-well tissue culture treated plate (Corning, cat#3764) in 20 µL of cRPMI. Cells were allowed to adhere for 6 hours at 37 °C and 5% $CO_2$. 20 µL of a serial dilution of the indicated antibody in cRPMI was then added to the cells and incubated overnight under the same conditions. The following day, 40 µL of Bright-Glo was added to each well and luciferase signal was read on a Perkin Elmer Envision plate reader.

For the IL-2 reporter assay, Jurkat cells were engineered in-house to express IL-2Rβ, IL-2Rγ, and a STAT5-luciferase reporter (Jurkat$^{\beta\gamma\text{-STAT5-Luc}}$). 20,000 cells in 20 µL of cRPMI were added to 20 µL of cRPMI containing serial dilutions of antibodies or recombinant IL-2 and incubated overnight at 37 °C and 5% $CO_2$. The following day, 40 µL of Bright-Glo reagent was added to each well and luciferase signal was quantified using a Perkin Elmer Envision plate reader. For IL-2 blocking experiments, cells were first coated with 1 µM of each monospecific anti-IL-2Rβ or anti-IL-2Rγ clone for 1 h prior to the addition of an IL-2 serial dilution. Schematics of each assay can be found in Fig. S4A-C.

## Primary cell assays

Purified primary human CD8 + T cells (cat#70027) or NK cells (cat#70036) were purchased from STEMCELL technologies. Each cell type was obtained from a different individual donor ($n = 1$) and isolated to >85% purity by STEMCELL technologies. CD8 + T cells were pre-

stimulated with a 1:1 ratio of CD3/CD28 T-Activator Dynabeads (Gibco, cat#11131D) at $1 \times 10^6$ cells/mL in cRPMI at 37 °C and 5% $CO_2$. After 48 hours the Dynabeads were magnetically separated from the cells and cells were allowed to rest overnight in cRPMI at 37 °C and 5% $CO_2$. 50 µl or cRPMI containing 25,000 cells was then added to 50 µL of cRPMI containing a serial dilution of the indicated mimetic antibody format or recombinant IL-2 and incubated at 37 °C and 5% $CO_2$ in white 96-well plates (Corning, cat#3917). After 48 hours, 100 µL of CellTiter-Glo 2.0 (Promega, cat#G9242) was added to each well and luciferase signal was read on a Perkin Elmer Envision plate reader. The same protocol was followed for NK cells with the exception of the pre-stimulation step.

## RNA-seq

Purified primary human CD8 + T cells (STEMCELL technologies, cat#70027) were pre-stimulated and rested as described above. $2 \times 10^6$ cells were plated in 2 mL of cRPMI in 6 well plates. Each well was treated in triplicate with 100 nM of the indicated mimetic antibody format or recombinant IL-2 and incubated at 37 °C and 5% $CO_2$ for 24 h. Cells were pelleted and RNA was extracted using an RNeasy mini kit (Qiagen, cat#74014).

Total RNA was quantified with Qubit RNA HS Assay Kit (Thermo-Fisher) and quality was assessed using RNA ScreenTape on 4200 TapeStation (Agilent Technologies). For sequencing library generation, the Truseq Stranded mRNA kit (Illumina) was used with an input of 100 ng of total RNA. Libraries were quantified with Qubit dsDNA HS Assay Kit (ThermoFisher) and the average library size was determined using D1000 ScreenTape on 4200 TapeStation (Agilent Technologies). Libraries were pooled and sequenced on NovaSeq 6000 (Illumina) to generate 30 million single-end 50-base pair reads for each sample.

RNA-sequencing data were analyzed using HTSeqGenie[72] in BioConductor[73] as follows: first, reads with low nucleotide qualities (70% of bases with quality <23) or matches to rRNA and adapter sequences were removed. The remaining reads were aligned to the human reference genome (human: GRCh38.p10) using GSNAP[74,75] version '2013-10-10-v2', allowing maximum of two mismatches per 75 base sequence (parameters: '-M 2 -n 10 -B 2 -i 1 -N 1 -w 200000 -E 1 --pairmax-rna=200000 --clip-overlap'). Transcript annotation was based on the Gencode genes data base (human: GENCODE 27). To quantify gene expression levels, the number of reads mapping unambiguously to the exons of each gene was calculated.

Differential expression was calculated using EdgeR[76] (version 3.40.1), grouping on replicates of each condition relative to the negative control condition (gD). Differentially expressed genes were determined using a Benjamini-Hochberg False Discovery Rate of 1%. The log2 fold-change of differentially expressed genes were normalized within each condition using sklearn.preprocessing.normalize[77] (scikit-learn version 1.0.1) before hierarchical clustering was performed using scipy.cluster.hierarchy[78] (SciPy version 1.7.3, parameters: method = 'ward').

## Statistics and reproducibility

Statistical analysis for specific experiments can be found within the relevant Methods subsection. Sample size was guided by assay throughput. Cell-based agonism assays had at least $n = 2$ independent wells along with wide-spanning titration curves. Each assay was repeated 2-4 times with robust reproducibility. All other experiments were performed once, guided by throughput as well as time and resource requirements. No statistical method was used to predetermine sample size. No data were excluded from the analysis. No experiments were randomized or blinded.

## Reporting summary

Further information on research design is available in the Nature Portfolio Reporting Summary linked to this article.

## Data availability

The data that support the findings of this study are available within the paper, its supplementary information files, the source data file, or at the following repositories with the specified accession codes. Source data are provided as a Source Data file. RNAseq data that support the findings of this study have been deposited in the Sequence Read Archive (SRA) with accession code GSE246602. Amino acid distributions in Fig. 1b were obtained from the abYsis database (http://www.abysis.org/). Protein structures shown in Fig. 1a, 4c were obtained from the Protein Data Bank with accession codes 2OQJ, 7LU9, 7L6M, and 2ERJ. Source data are provided in this paper.

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

## Acknowledgements

The authors thank Michael Dillon, Norbert Leclair, Art Orjalo, and Ryan Kelly for their technical contributions. Funding for this work was pro-vided internally by Genentech Inc.

## Author contributions

Conceptualization: M.G.R., B.L., J.T.S., G.A.L. Methodology: M.G.R., B.L., Z.B.K., D.L., Y.Y., E.S.D., J.B., G.N., J.T.S. Investigation: M.G.R., B.L., Z.B.K., D.L., Y.Y., E.S.D., C.W.K., P.S., J.B., I.K., H.D., F.F., M.L. Visualization: M.G.R., B.L., Z.B.K., D.L., E.S.D. Supervision: A.S.S., J.T.S., G.A.L. Writing: M.G.R., B.L., Z.B.K., D.L., E.S.D., J.T.S., G.A.L.

## Competing interests

All authors are current or former employees of Genentech, mem-ber of the Roche Group, and shareholders in Roche. This study was supported by internal Genentech funds, and the funders had no role in study design, data collection and analysis, decision to

publish, or preparation of the manuscript. The authors declare no other competing interests.
