## [Peer Review File · Nature Communications]

REVIEWER COMMENTS

Reviewer #1 (Remarks to the Author):

Romei et al. described an elegant approach to using a conformational tuning approach to engineer IgG-like antibodies with enhanced therapeutic potential. This approach used mutation-guided design informed by dimerized antibodies previously found to target glycans on surface protein of HIV, Coronavirus spike and yeast antigens. In this study, Romei and colleagues demonstrated that Fab-dimerized forms of anti-OX40 and anti-IL2 antibodies might have improved functions, based on antibody agonism readout in the culture assays. Overall, high enthusiasm for this study, but few suggestions to improve understanding for a general nature communications audience and better understanding of the biology and therapeutic role of these antibodies:

1. What is the target for the monomeric versus dimerized anti-OX40 and anti-IL2 antibodies?
2. Does dimerization only maintain or also expand antigenicity?
3. Are OX40 and IL2R glycosylated, and possibly explain the epitope targeted more effectively by the dimerized antibodies?
4. As shown in figure S2, the authors were able to nicely show separation of the monomers and dimers within iAbaff2 via SEC. Using this approach; can they demonstrate that the improved agonism observed by the iAbdx/ iAbaffa/iAbaff2 was due to primarily to the dimeric portions of the antibodies?
5. Would separating monomer and dimer species per iAb be difficult to achieve given the reversible dimer formation of iAbs as described? Is the reversible dimer formation a consequence of assay conditions or biology of these antibodies?
6. Williams et al (source of DH851 and DH898) mentioned difficulty separating these species, which may be a success of the current study. Something for authors to address, if possible.
7. Are there concerns about acquired polyreactivity profiles for dimerized anti-OX40 and anti-IL2 antibodies? If the goal is for use as therapeutic agents, this should be addressed.
8. For the human primary cells as source of NK and CD8 T cells, more details are needed in the methods and figure legend (Figure 5) to get a better idea of the number of individuals from whom PBMCs were obtained, and number and quality of NK and CD8 T cells analyzed per assay, etc. This will provide a better understanding on the impact of the agonist on primary cells that may translate to humans.
9. Introduction and discussion can be revised to provide better translational capacity of these dimerized antibodies. As written, these sections are dense with protein chemistry and lack clear purpose for the clinical applicability or next steps for these antibodies.

10. What is known about previous OX40 agonists and whether these dimerized iAb (anti-OX40) antibodies provide an improvement? The answers to these questions may also be used to improve the introduction and discussion.

11. Graphical images describing some of the assays will help readership. No clear methods for anti-OX40 agonism assay; this should be included.

Minor comments:

1. Figure 2; which anti-OX40 Ab clone (WT) was shown in the 2D class average image? Not clear from the text or legend.

2. Lines 450-452; I don't understand this sentence.

Reviewer #2 (Remarks to the Author):

Summary:

In this paper the authors develop a method of modifying existing monoclonal antibodies into a format that allows for receptor agonism. The engineering results in the non-covalent association of the two Fab fragments of an igg into a format they call an i-shaped antibody (iAb). Their design is based on reports of antibodies that naturally assume this format. They report 3 designs based on amino acid substitutions made in the VH domain of the heavy chain to enable iAb formation. However, they report the iAb form to exist in equilibrium with the traditional Y-shaped form of each construct. The best design efficiently induced OX40 agonism when applied to many different antibodies targeting OX40. They showed the best overall agonism for a construct in F(ab')₂ format in figure 2, but this format was not investigated further. They showed superior agonism in their format for anti-OX40 antibodies compared with the same in the previously described contorsbody format. They found somewhat reduced agonism and internalization of an anti-OX40 IgG in the iAb format compared with the hexameric format. Next, the authors generated a panel of antibodies against IL-2RB and IL-2RBG by yeast display. They used the identified antibodies to generate a panel of bispecifics in both iAb and contorsbody formats. Testing revealed a greater fraction of antibodies in contorsbody format to induce IL-2 agonism in their Jurkat reporter cell line. The best contorsbody bispecific also seemed to slightly outperform their iAbs at inducing NK and CD8 proliferation.

Major comments:

1. Lines 208-213: The authors need to determine the percentage of the F(ab')₂ that is present in iAb form (i-shaped conformation) than Y-shaped form, perhaps as done in Figure 2A. Also, were the iAbs tested in hexameric or any other format?

2. Figure 2E: The F(ab')₂ format appears to be superior for the antibody tested in this panel. This needs to be confirmed by testing it with other anti-OX40 antibodies and with the bispecific antibodies against IL-2R.
3. Figure 3: The iAb also should be tested in hexameric format
4. Figure 3: It's not clear how much agonism, internalization or clustering should be expected, or is ideal. OX40L need to be used as a positive control in this experiment.
5. In Figure 5: was there any confirmation of "iAb formation" for the bispecifics tested? Could lack of iAb formation/stability explain the lack of functionality of so many of the tested constructs (in addition to the explanation given in lines 436-448 of the discussion)? The authors need to determine, or indicate the percentage of each construct that is present in the iAb i-shaped conformation.
6. Why is the response for B10/G25 iAb apparently stronger in Figure 5B than in Figure 5A at the 100 nM concentration?
7. For figure 5E: can the authors offer some comment on the differences in induced expression between the iAbs tested?
8. Figure S2: Since many of the iBaff2 constructs were dimers, was a similar trend observed with the iBaff1 constructs? The authors need to show the SEC data for iBaff1 and iAbdx constructs corresponding with those shown for the iBaff2 constructs.

Minor comments:

1. For all figures: Missing error bars for several figures. Please indicate numbers data points used to generate error bars, and the number of times each experiment was done.
2. Did not mention that some mAbs can function as agonists (there are several examples)(1).
3. For Figure 5C what does the fold increase refer to?
4. Line 75: Use of language: "mimetic" should be "mimic"
5. Line 236: should read: "empirical in nature"
6. Line 366: "activated" should be spelled "activated"
7. Figure 5 caption: "IL-2RG/IL-2RB" should be written as "IL2-Rγ/IL-2Rβ" for consistency.
8. The rest of the manuscript should be rechecked for language and consistency.

1. Vonderheide RH. 2020. CD40 Agonist Antibodies in Cancer Immunotherapy. Annual Review of Medicine 71:47-58.

Reviewer #3 (Remarks to the Author):

The authors describe exploitation of naturally occurring Fab/Fab intramolecular interactions to drive agonism of TNFR and cytokine receptors. The protein engineering and structural characterization are executed with a high degree of technical proficiency, and are convincing. There remain some underexplored issues comparing the maximum activity of these constructs by contrast with natural ligands. These differences are important, since the exploration of the i-body format is motivated by the developability and pharmacological advantages of the IgG format, which unfortunately has lower maximum activity than the less-developable and shorter PK-retention F(ab2)' construct (Figure 2E). However, as an intriguing update on a novel topological IgG construct, this paper is interesting and worthy of publication.

The comparison across a wide variety of receptors (CD40, 4-1BB, DR4, DR5, Figs 2F-1) effectively demonstrate the generality of the I-body format, but leave some doubt as to the achievable maximum activity. For CD40, this seems to fluctuate between 2-8x an untreated control, 4-1BB 2-4x. Since these activities are expressed normalized against an untreated sample, it is not clear how this would compare quantitatively to the natural ligands. Those positive controls (CD40L, 4-1BBL) should be performed and reported for comparison.

The paper presents a comprehensive comparison of different i-body bispecific anti-IL-2R antibodies, testing 6x8 bispecifics in 3 conformations each. The Contorsbody conformation outperformed others, yielding more agonistic combinations. Interestingly, the natural ligand IL-2 showed similar maximal activity to the active bispecific conformations (Figure 5 B, D), but was more potent (Figure 5B). However, IL-2 had a more consistent and high impact on gene expression than any antibody-based agonists (Figure 5E). This limitation of the antibody agonists warrants more emphasis as it raises questions about the feasibility of replacing natural cytokines with such antibodies.

REVIEWER COMMENTS

Reviewer #1 (Remarks to the Author):

Romei et al. described an elegant approach to using a conformational tuning approach to engineer IgG-like antibodies with enhanced therapeutic potential. This approach used mutation-guided design informed by dimerized antibodies previously found to target glycans on surface protein of HIV, Coronavirus spike and yeast antigens. In this study, Romei and colleagues demonstrated that Fab-dimerized forms of anti-OX40 and anti-IL2 antibodies might have improved functions, based on antibody agonism readout in the culture assays. Overall, high enthusiasm for this study, but few suggestions to improve understanding for a general nature communications audience and better understanding of the biology and therapeutic role of these antibodies:

We thank the Reviewer for their time and helpful comments.

1. What is the target for the monomeric versus dimerized anti-OX40 and anti-IL2 antibodies?

With the exception of a single CDRH2 mutation in iAb_{dx}, the CDRs of all 3 engineered versions (iAb_{dx}, iAb_{aff1}, and iAb_{aff2}) are identical to the parental Abs, and thus have the same target specificity. Thus Fab-dimerized and monomeric anti-OX40 antibodies all bind OX40, and Fab-dimerized and monomeric IL-2R β and IL-2R γ antibodies bind IL-2R β and IL-2R γ , respectively. A sentence has been added to the Results section (lines 139-141) to clarify this. In addition, Fig. 2C and the accompanying paragraph in the Results section (lines 184-194) demonstrate that at least for iAb_{aff1} the engineered versions do not alter solution or cell-based affinity for OX40.

2. Does dimerization only maintain or also expand antigenicity?

We are not sure we understand what the Reviewer is asking, but we will respond with our interpretation. Antigenicity to us refers to an aspect of target recognition, in this context the ability of a target antigen (OX40, CD40, 4-1BB, DR4, DR5, IL-2R β , or IL-2R γ) to be specifically recognized by antibodies. With that definition, we expect dimerization to maintain but not expand antigenicity.

3. Are OX40 and IL2R glycosylated, and possibly explain the epitope targeted more effectively by the dimerized antibodies?

The extracellular domain of OX40 is glycosylated. Structures have been solved for two of the anti-OX40 antibodies 1A7 and 3C8 (Yang et al., 2019, mAbs, which has been referenced) indicating that they do not bind epitopes that contain glycan. So at least in those two cases the increase in activity cannot be attributed to glycan binding. We do not know about the epitopes of the other anti-OX40 antibodies, nor the IL-2R β or IL-2R γ antibodies.

We appreciate the Reviewer's insight, that perhaps the increased activity of the engineered versions could be related to glycan-repeat motifs for which enhanced activity was observed in the work in which these interactions were discovered (Calarese et al., Science 2003; Williams et al., 2021, Cell). A distinction of those works from the present set of antibodies is that, in contrast to those pathogenic glycan motifs, the glycosylation patterns on the receptors tested in the present study are generally not repeat motifs, and thus we would not expect the same mechanistic principle to apply.

4. As shown in figure S2, the authors were able to nicely show separation of the monomers and dimers within iAb_{aff2} via SEC. Using this approach; can they demonstrate that the improved agonism observed by the iAb_{dx}/ iAb_{affa}/iAb_{aff2} was due to primarily to the dimeric portions of the antibodies?

The electron microscopy data establish that iAb_{aff2} forms inter-molecular dimers (Fig. 2 A). In contrast, iAb_{dx} and iAb_{aff1} only show monomeric species. The sedimentation data show that 3C8 Ab_{aff1} Fab does form weak reversible inter-molecular dimers in solution with an experimentally determined $KD = 6.8 \mu M$ that is higher than the concentrations used in our assays. That said, the activity of the 3C8 iAb_{aff1} Fab, albeit very weak, does indicate that inter-molecular dimer can contribute to activity. The over 100-fold greater activity of the iAb_{aff1} IgG relative to iAb_{aff1} Fab indicates that the improved agonism of the IgG version cannot be attributed primarily to a putative intermolecular dimer component.

This comment together with comments 5 and 6 below make evident that more discussion is needed on this aspect of our work. We thank the Reviewer for highlighting this gap. A new paragraph has been added to the Discussion (lines 460-485) that provides readers greater clarity and discussion on the monomer / dimer aspect of the engineered formats.

5. Would separating monomer and dimer species per iAb be difficult to achieve given the reversible dimer formation of iAbs as described? Is the reversible dimer formation a consequence of assay conditions or biology of these antibodies?

Yes, separation would not be possible since the populations are in equilibrium. Our interpretation is that reversible dimerization is intrinsic to the molecules and not a consequence of assay conditions or biology; more specifically, reversible dimer formation is due to the nature of the interface and the fact that it is noncovalent with relatively weak affinity. Comments on this aspect have been included in the new paragraph added to the Discussion as mentioned above.

6. Williams et al (source of DH851 and DH898) mentioned difficulty separating these species, which may be a success of the current study. Something for authors to address, if possible.

While we were able to observe dimer formation of iAb_{aff2} by analytical SEC, the reversible nature of the interaction precludes preparative separation of monomer and

dimer or higher order species. Comments on this aspect have been included in the new paragraph added to the Discussion as mentioned above.

7. Are there concerns about acquired polyreactivity profiles for dimerized anti-OX40 and anti-IL2 antibodies? If the goal is for use as therapeutic agents, this should be addressed.

Each antibody clone is specific for its antigen, and we do not have data to suggest that the engineering changes polyreactivity of the parent clone. While we don't expect there to be, the Reviewer raises a fair point about this in the context of application as therapeutic agents. This comment together with comments 9 and 10 below make evident that more discussion is needed on therapeutic translational considerations and next steps. We thank the Reviewer for highlighting this gap. A new paragraph has been added to the Discussion (lines 517-541) that acknowledges that further study of many aspects for development is needed. Regarding polyspecificity, references are made to relevant papers that describe assays to test for this and other developability properties.

8. For the human primary cells as source of NK and CD8 T cells, more details are needed in the methods and figure legend (Figure 5) to get a better idea of the number of individuals from whom PBMCs were obtained, and number and quality of NK and CD8 T cells analyzed per assay, etc. This will provide a better understanding on the impact of the agonist on primary cells that may translate to humans.

More details have been added to both the Methods (lines 822-823) and figure legend for Figure 5.

9. Introduction and discussion can be revised to provide better translational capacity of these dimerized antibodies. As written, these sections are dense with protein chemistry and lack clear purpose for the clinical applicability or next steps for these antibodies.

We thank the Reviewer for noting this gap. The Introduction section does have discussion of translation of antibodies, but without yet having introduced the results we decided a discussion of translation of the iAb engineered versions was best done in the Discussion. As mentioned above, we have added a new paragraph to the Discussion to discuss translation and clinical applicability.

10. What is known about previous OX40 agonists and whether these dimerized iAb (anti-OX40) antibodies provide an improvement? The answers to these questions may also be used to improve the introduction and discussion.

As mentioned above, we have added a new paragraph to the Discussion to discuss translation and clinical applicability. In that paragraph we elaborate on clinical development of previous agonists, including OX40, and how the engineered iAbs may provide an improvement.

11. Graphical images describing some of the assays will help readership. No clear methods for anti-OX40 agonism assay; this should be included.

A new supplementary figure, Fig. S4, has been added that provides graphical images of the cell-based assays utilized in this manuscript. A reference was made to the previously described OX40 assay, but we have added additional text to the methods (lines 795-801) to include some of those details in the present manuscript.

Minor comments:

1. Figure 2; which anti-OX40 Ab clone (WT) was shown in the 2D class average image? Not clear from the text or legend.

3C8 was used as a representative Ab clone for all versions in the images. We have added a sentence to clarify this in the legend for Figure 2A.

2. Lines 450-452; I don't understand this sentence.

Lines 450-452 in the submitted manuscript are below:

While this study demonstrates proof-of-concept for the use of the conformationally constrained iAb platform for agonist antibody design and generates intriguing hypotheses about receptor biology, unknowns remain.

We infer that the Reviewer takes issue with the phrase about hypotheses about receptor biology. We agree that this is vague and unnecessary and have deleted it.

Reviewer #2 (Remarks to the Author):

Summary:

In this paper the authors develop a method of modifying existing monoclonal antibodies into a format that allows for receptor agonism. The engineering results in the non-covalent association of the two Fab fragments of an igg into a format they call an i-shaped antibody (iAb). Their design is based on reports of antibodies that naturally assume this format. They report 3 designs based on amino acid substitutions made in the VH domain of the heavy chain to enable iAb formation. However, they report the iAb form to exist in equilibrium with the traditional Y-shaped form of each construct. The best design efficiently induced OX40 agonism when applied to many different antibodies targeting OX40. They showed the best overall agonism for a construct in F(ab')₂ format in figure 2, but this format was not investigated further. They showed superior agonism in their format for anti-OX40 antibodies compared with the same in the previously described contorsbody format. They found somewhat reduced agonism and internalization of an anti-OX40 IgG in the iAb format compared with the hexameric format. Next, the authors generated a panel of antibodies against IL-2RB and IL-2RBG by yeast display. They used the identified antibodies to generate a panel of bispecifics

in both iAb and contorsbody formats. Testing revealed a greater fraction of antibodies in contorsbody format to induce IL-2 agonism in their Jurkat reporter cell line. The best contorsbody bispecific also seemed to slightly outperform their iAbs at inducing NK and CD8 proliferation.

We thank the Reviewer for their careful read of our manuscript and insightful comments.

Major comments:

1. Lines 208-213: The authors need to determine the percentage of the F(ab')₂ that is present in iAb form (i-shaped conformation) than Y-shaped form, perhaps as done in Figure 2A. Also, were the iAbs tested in hexameric or any other format?

Per the reviewer's request, we analyzed the anti-OX40 antibody, 3C8, with the iAb_{aff1} residue set in the F(ab')₂ format by negative stain electron microscopy. We have included the images of the 2D classes in the SI as Fig. S9 and have added some accompanying discussion about the results in the main text (lines 217-222).

Interestingly, we observe no i-shaped conformation of the F(ab')₂ in the negative stain 2D classes. This is a surprising result, since we observe potent activity of the F(ab')₂ format in the OX40 agonism assay, and the Fab-Fab interaction intuitively should be independent of the Fc. However, there is precedent for antibody formats that appear monomeric through biophysical characterization to induce clustering-based agonism activity. For example, the iAb_{aff1} Fab shown in Fig. 2E has agonism activity despite being a monomeric Fab at relevant assay concentrations. Analytical ultracentrifugation analysis revealed an intermolecular homodimerization affinity for the iAb_{aff1} Fab of 6.8 μM (see Fig. S8 and Table S1), and the maximum assay concentration was <1 μM. A second example explored in the literature involves Fc hexamerization mutations. Zhang et al. and Wang et al. (Refs. 67 and 68) show that a single mutation in the Fc induces potent agonism for an anti-OX40 antibody, but SEC analysis confirms that the antibody is monomeric in solution. To achieve a stable hexamer in solution, three Fc mutations are required. Collectively these results suggest that avidity of antibody-receptor interactions on the cell surface can be sufficient to induce intermolecular antibody interactions mediated by the iAb_{aff1} mutations.

Addressing the final question, we did not combine the iAb mutations with the Fc hexamer mutations to create a hexameric iAb. This combination of engineering tools is a good idea, and agonism activity may be additive or synergistic. However, the experiment to explore the suggested format is outside the scope of the present study. Our goal was to develop the most simplistic, therapeutically developable platform possible that will induce agonism activity. A major benefit of the iAb format is the small number of mutations required and the structural similarity to a standard IgG. Hexameric therapeutic antibodies have the potential to introduce additional risk such as developability and PK.

2. Figure 2E: The F(ab')₂ format appears to be superior for the antibody tested in this

panel. This needs to be confirmed by testing it with other anti-OX40 antibodies and with the bispecific antibodies against IL-2R.

The F(ab')₂ does have greater maximum activity based on the data in Fig. 2E, but it does not have greater potency than the iAb IgG based on EC₅₀ values. Additionally, with our aim being drug discovery, the F(ab')₂ format is suboptimal to IgG with respect to manufacturing and PK, and thus any apparent greater activity in vitro would not translate in vivo due to its rapid clearance. Echoing the response to the comment above, our focus was on developing an engineering tool with the least risk to therapeutic development, so we did not pursue the F(ab')₂ further. Testing the F(ab')₂ format in all other TNFRSF antibodies and all bispecific IL-2R antibodies would not be a trivial task due to the large number of antibodies and the resources required for bispecific assembly. While we agree the experiment would be interesting, given the demanding resources yet non-impact to the paper's conclusions, we believe such an experiment is outside the scope of the present study.

This comment together with the comment above make evident that more discussion is needed on our de-emphasis of the F(ab')₂ and our deeper characterization of the IgG iAb moving forward. We thank the Reviewer for highlighting this gap. We wish to emphasize that in our view the more important result from these experiments is that the iAbs are able to activate receptor without crosslinking, in contrast to the inert WT control as shown in the figures. Our wish was to advance that capability in an IgG-like format so as not to add molecule risk or unnecessary liabilities. We have added a brief statement to the relevant Results section (lines 219-225), and a new paragraph to the Discussion (lines 517-541) that provides readers with greater context and rationale for our focus on the IgG version of the iAb.

3. Figure 3: The iAb also should be tested in hexameric format

We addressed this point in our response to the first comment above. In short, our goal was to identify a developable platform with the most promise and simplicity to advance as a therapeutic. Testing an iAb hexamer would be an interesting pursuit, but it is outside the scope of this study and would not change our conclusions.

4. Figure 3: It's not clear how much agonism, internalization or clustering should be expected, or is ideal. OX40L need to be used as a positive control in this experiment.

The iAb approach to receptor agonism is new, and rather than project an expectation on it relative to ligand, we felt best to just share the data. We agree that comparison with ligand is important and thank the Reviewer for highlighting this gap. To address this point we have added a supplemental figure comparing the activity of OX40L to that of our iAb format (Fig. S5) and referenced this figure in the results section on line 180. Technical limitations precluded internalization or clustering experiments in a directly comparable and relevant manner.

5. In Figure 5: was there any confirmation of "iAb formation" for the bispecifics tested?

Could lack of iAb formation/stability explain the lack of functionality of so many of the tested constructs (in addition to the explanation given in lines 436-448 of the discussion)? The authors need to determine, or indicate the percentage of each construct that is present in the iAb i-shaped conformation.

The reviewer brings up a great point about confirmation of iAb formation that we have spent much time and resources trying to address. In this work, there are 40 anti-TNFRSF antibodies and 48 bispecific anti-IL-2R antibodies that we have produced as iAbs and tested in activity assays, most of which have been produced and tested in multiple formats. Analysis of these antibodies to determine percent iAb formation in solution requires at least a medium- to high-throughput quantitative assay. This assay cannot require an excessive quantity of material, especially considering the limitations of our bispecific assembly process. Electron microscopy is a low throughput technique, and with high instrument demand and project gates we simply are not resourced to perform negative stain electron microscopy on all 88 iAb samples (not counting additional iAb residue sets, alternative molecule formats, and control antibodies). Aware of these limitations, we attempted a wide range of additional biophysical techniques in an effort to reliably quantify iAb formation in solution. Experimental approaches included analytical ultracentrifugation, dynamic light scattering, fluorescence anisotropy, and several chromatographic methods. None provided reliable quantification of iAb formation in a medium to high-throughput manner for a variety of technical reasons.

We don't disagree with the Reviewer's wish for more insight into this aspect, but it's just not practical at the scale of all our combinations. As a compromise, we chose one bispecific antibody that demonstrated IL-2 agonism activity (B10/G28) and one that was inactive (B09/G28) but with the same and IL-2R γ clone, and we analyzed those two representative samples with negative stain electron microscopy to determine extent of iAb formation. We have added images of the 2D classes to the SI as Fig. S13, and we have included discussion in the results section (lines 352-359). In brief, both bispecific antibodies with the iAb_{aff1} residue set adopt the i-shaped conformation as determined by negative stain. Both bispecific iAbs have a distribution of conformations with the B10/G28 and B09/G28 iAbs containing 39% and 59% i-shaped particles, respectively, with the remaining particles adopting the traditional Y-shaped conformation. Note that a variety of factors likely contribute to these conformational distributions, so the exact distribution on the cell surface could be different than that in solution.

The main conclusion from this new data is that the difference in IL-2 agonism activity from the bispecific iAbs does not stem from differences in conformation, or more specifically, the presence or absence of the i-shaped conformation. They are roughly equivalent in that aspect. Instead, these activity differences may be due to the specific epitope and/or affinity of each antibody within the bispecific, or potentially the interdependent characteristics of the antibody pair. Perhaps the epitopes must be a certain distance apart from each other on the receptor complex, or the binding orientations of the Fabs could play a role. In this example, the two bispecific iAbs differ in the anti-IL β antibody clone. Fig. 4C suggests B09 and B10 have similar epitopes, but we do not have any information on the orientation of Fab binding or the paratope of the

Fab. Fig. 4B shows that B09 and B10 differ in binding affinity and cell surface binding, which could also contribute to the activity differences.

6. Why is the response for B10/G25 iAb apparently stronger in Figure 5B than in Figure 5A at the 100 nM concentration?

This discrepancy is due to the fact that the assays in Fig. 5A and 5B were performed on different days, and the absolute value of the signal for the luciferase readout can vary based on a variety of factors across assay repeats. The goal of the experiment in 5A was to screen a larger test set for hits that we then explored more deeply in 5B. Importantly, the relative value of the signal among samples and the trends within a single assay are consistent. For example, the data at 100 nM of B10/G25 is slightly lower than that of B10/G28 in both Figs. 5A and 5B. Thus, despite minor differences between experiments, the screen achieved its function.

7. For figure 5E: can the authors offer some comment on the differences in induced expression between the iAbs tested?

The induced expression levels between the bispecific iAbs B10/G25 and B10/G28 follow the same trends as the Jurkat and primary cell agonism assay data. The B10/G25 iAb has a lower maximum activity and EC_{50} value compared to the B10/G28 iAb (Figs. 5A, 5B, and 5D). Consistent with those trends, Fig. 5E shows that the B10/G25 iAb has lower induced expression of many of the up-regulated genes compared to B10/G28 (e.g., TRPM2, TNF, SLC34A1, BCL2L14, ETV7, TNFRSF8, and OSM). These data in combination with the rest of the data in Fig. 5 suggest that B10/G28 is a better IL-2 agonist than B10/G25. To highlight these observations, we have added text to the relevant results section (lines 393-397).

8. Figure S2: Since many of the iAbaff2 constructs were dimers, was a similar trend observed with the iAbaff1 constructs? The authors need to show the SEC data for iAbaff1 and iAbdx constructs corresponding with those shown for the iAbaff2 constructs.

To address this comment, we have added a new figure in the SI as Fig. S2 that shows SEC data comparing WT, iAb_{aff1}, and iAb_{aff2} for each anti-OX40 clone. In brief, all iAb_{aff1} antibodies are monomeric. There are small differences in elution time between the WT and iAb_{aff1} antibodies for each anti-OX40 clone, with a general trend of a shift to longer elution times for the iAb_{aff1} antibodies, which may reflect the more compact shape as well as increased hydrophobicity of the protein. Most of the iAb mutations introduce hydrophobic residues in the Fabs, and the samples exist in an equilibrium of i-shaped and Y-shaped conformations. Yet, the data clearly show large elution time shifts for most of the iAb_{aff2} antibodies which, coupled with SEC-MALS in Fig. S3, indicates dimer formation. We have added text in the relevant Results section to clarify this point and reference the new SI figure (lines 155-156 and 158-159).

The additional SI figure does not contain the iAb_{dx} chromatograms. Due to the nature of these antibodies and the induced domain exchange effect, these antibodies formed a combination of monomer and dimer after expression. As shown by previous literature on the 2G12 domain-exchanged antibody, the monomer and dimer populations are not in equilibrium and do not exchange. Therefore, we simply used SEC to purify the monomer for all iAb_{dx} antibodies and discarded all dimers and higher order species.

Minor comments:

1. For all figures: Missing error bars for several figures. Please indicate numbers data points used to generate error bars, and the number of times each experiment was done.

We have added the number of replicates within figure legends for each experiment where applicable. In many cases, the error bars are smaller than the size of the point, so they are not always visually evident for all data points in the figure.

2. Did not mention that some mAbs can function as agonists (there are several examples)(1).

We discuss the existence of mAb agonists while highlighting the pitfalls in their discovery in the first paragraph of our discussion. We have modified the text in that section and added the suggested citation. In addition, we have added a new paragraph to the Discussion to discuss translation and clinical applicability. In that paragraph, we elaborate on clinical development of previous agonists and how the engineered iAbs may provide an improvement (lines 517-541).

3. For Figure 5C what does the fold increase refer to?

The y-axis in Fig. 5C refers to the fold-change of absorbance signal over control that lacks antibody sample. This axis is explained in the last sentence of the "Bridging ELISA" methods section. For better ease for readers, we have added text to the legend for 5C to clarify the y-axis.

4. Line 75: Use of language: "mimetic" should be "mimic"

We made the correction in the text.

5. Line 236: should read: "empirical in nature"

We made the correction in the text.

6. Line 366: "activiated" should be spelled "activated"

We made the correction in the text.

7. Figure 5 caption: “IL-2RG/IL-2RB” should be written as “IL2-R γ /IL-2R β ” for consistency.

We made the correction in the text.

8. The rest of the manuscript should be rechecked for language and consistency.

We thank the reviewer for the suggestion, and we have thoroughly proof-read the manuscript.

1. Vonderheide RH. 2020. CD40 Agonist Antibodies in Cancer Immunotherapy. Annual Review of Medicine 71:47-58.

Reviewer #3 (Remarks to the Author):

The authors describe exploitation of naturally occurring Fab/Fab intramolecular interactions to drive agonism of TNFR and cytokine receptors. The protein engineering and structural characterization are executed with a high degree of technical proficiency, and are convincing. There remain some underexplored issues comparing the maximum activity of these constructs by contrast with natural ligands. These differences are important, since the exploration of the i-body format is motivated by the developability and pharmacological advantages of the IgG format, which unfortunately has lower maximum activity than the less-developable and shorter PK-retention F(ab $'_2$)' construct (Figure 2E). However, as an intriguing update on a novel topological IgG construct, this paper is interesting and worthy of publication.

We thank the Reviewer for their careful read of our manuscript and their insightful comments. We agree with the view that developability and fast clearance would make the F(ab $'_2$) suboptimal to an IgG as a therapeutic format, which is why despite its apparent higher level of activity (albeit equivalent potency) we chose to de-emphasize the F(ab $'_2$) and more deeply characterize the IgG iAb moving forward. We also wish to emphasize that in our view the more important result from these experiments is that the iAbs are able to activate receptor without crosslinking, in contrast to inert WT control as shown in the figures. Our wish was to advance that capability in an IgG-like format so as not to add molecule risk or unnecessary liabilities. We have added a brief statement to the relevant Results section (lines 219-225), and a new paragraph to the Discussion (lines 517-541) that provides readers greater context and rationale for our focus on the IgG version of the iAb.

The comparison across a wide variety of receptors (CD40, 4-1BB, DR4, DR5, Figs 2F-1) effectively demonstrate the generality of the I-body format, but leave some doubt as to the achievable maximum activity. For CD40, this seems to fluctuate between 2-8x an untreated control, 4-1BB 2-4x. Since these activities are expressed normalized against an untreated sample, it is not clear how this would compare quantitatively to the natural

ligands. Those positive controls (CD40L, 4-1BBL) should be performed and reported for comparison.

We thank the Reviewer for the comment regarding comparisons to the activity of native ligands, which was also raised by Reviewer #2. As discussed above in our response to Reviewer #2, we have addressed this comment by adding new data that compares the activity of our iAb format to that of the native ligands for 3 receptors (OX40, CD40, and 4-1BB). In each case, the activity of the iAb is comparable to or enhanced relative to that of the native ligand (Fig. S5). We have also cited this result in the text on line 180 for OX40 and 245-247 for CD40 and 4-1BB.

The paper presents a comprehensive comparison of different i-body bispecific anti-IL-2R antibodies, testing 6x8 bispecifics in 3 conformations each. The Contorsbody conformation outperformed others, yielding more agonistic combinations. Interestingly, the natural ligand IL-2 showed similar maximal activity to the active bispecific conformations (Figure 5 B, D), but was more potent (Figure 5B). However, IL-2 had a more consistent and high impact on gene expression than any antibody-based agonists (Figure 5E). This limitation of the antibody agonists warrants more emphasis as it raises questions about the feasibility of replacing natural cytokines with such antibodies.

These are excellent points. To clarify, IL-2 had both similar maximal activity and potency as the iAbs in both the Jurkat reporter and against NK cells, but showed enhanced potency against CD8+ T cells. Our interpretation is that this is likely due to the expression of IL-2R α (CD25) on this particular cell type, a point we address in the text on lines 384-386. While the benefit of CD25-independence could be debated (beyond the scope of the present study), we suggest that this result at a minimum illustrates that an Ab approach can have the potential to offer different selectivity relative to a native cytokine. Regarding the higher impact on gene expression seen for IL-2, several factors could contribute to this observation, including IL-2R α binding as this experiment was performed using primary CD8+ T cells. Alternatively, or in addition, the difference may be a consequence of differing receptor geometries that result in altered signaling upon binding by native ligand vs antibody-based agonists. Perhaps screening of additional bispecific combinations at the transcriptional level could lead to an antibody-based agonist with more similarity to the native ligand. That being said, the gain-of-function resulting simply from altering the conformation of the antibody is quite striking (i.e. WT vs iAb/contorsbody), and unbiased hierarchical clustering suggests that the transcriptional signatures of the iAbs and contorsbodies are more similar to that of IL-2 than their WT counterparts. Regardless, we infer that more commentary is needed on this, and to address this important point we have strengthened our discussion of the RNAseq analysis to further highlight similarities and differences in the results (lines 393-397).

REVIEWER COMMENTS

Reviewer #1 (Remarks to the Author):

I thank the authors for their responses to my comments and questions. The revisions have addressed most of my concerns in the paper, but a few remain.

First, Fab dimerization of the antibodies that formed the basis for the dimerization design in this paper was reported to be beneficial for glycan-based epitope recognition. While the authors commented that OX40 was glycosylated, they did not provide sufficient evidence that can rule out glycan-dependent binding by the agonist designed in this paper. This limitation should be outlined or discussed in the manuscript.

Second, if the dimerization benefit for the antibodies studied is simply improving affinity for the same epitope, then this should be clearly outlined in the text and supported by a main text figure. Figure S5 showing the comparisons of the i-shaped antibodies to the natural ligands for the receptors studied would be more appropriate as a main text figure.

Third, in regard to the applicability of using these new agonists therapeutically, the authors did not indicate the number of individuals from whom primary cells were obtained. Instead, the number of replicate assays were shown. For example, on line 1155, there should be "N=X" next to individuals.

Fourth, the introduction and discussion could benefit from revisions to improve clarity and impact for translation of these results to basic/translational researchers. As is, it is written for a targeted audience and may be dense for nature communications readership.

Reviewer #2 (Remarks to the Author):

I am satisfied with the author responses to my comments (and the comments of the other reviewers).

Reviewer #3 (Remarks to the Author):

The revisions satisfactorily address this reviewer's critiques.

Reviewer #1 (Remarks to the Author):

I thank the authors for their responses to my comments and questions. The revisions have addressed most of my concerns in the paper, but a few remain.

First, Fab dimerization of the antibodies that formed the basis for the dimerization design in this paper was reported to be beneficial for glycan-based epitope recognition. While the authors commented that OX40 was glycosylated, they did not provide sufficient evidence that can rule out glycan-dependent binding by the agonist designed in this paper. This limitation should be outlined or discussed in the manuscript.

While the reviewer is correct that the Fab dimerizing antibody interface enabling agonist activity in this study was originally found to enhance glycan binding, there is no reason that the engineered antibodies in this manuscript should similarly utilize glycan binding, nor is there any evidence of it. We offer several reasons why we believe glycan binding is of minor importance or irrelevant to the present work.

- 1) The glycan reactive antibodies that the reviewer references were specifically isolated from subjects immunized or infected with HIV, and the surface of this virus consists of viral envelope proteins where repeated glycans comprise over 50% of the mass of the antigen. The antibodies of interest were then filtered based on glycan binding. In other words, the Fabs were specific for glycans, not protein motifs. Here, we utilized multiple panels of antibodies that bind a diverse set of epitopes across multiple targets that were not a priori selected for glycan binding. In fact, for the antibodies that we have epitope information, none binds glycans.
- 2) Figures 2D and 2E (formerly 2C and 2D) show that there is no change in binding affinity upon engraftment of the iAb-inducing residues, suggesting that the interaction of the antibodies with their cognate antigens is not altered. This is in contrast to the previous studies where Fab dimerization in the context of glycan-reactive antibodies acquired increases in affinity and/or avidity.
- 3) Previously reported structural evidence suggests that the Fab-dimerized conformation generates an additional glycan binding site between the two Fabs. Again, this is likely due to the fact that the Fabs themselves bind repeat glycan motifs, and there is an avidity effect with the density of glycans on the surface of the virus. In our work, there is no reason to believe that bringing together Fabs inherently creates a glycan binding site. The residues mutated in our antibody panels to induce iAb formation, especially for iAb_{aff1} and iAb_{aff2}, are buried within the newly formed interface between the Fabs. With the diverse panels of protein antigen antibodies we explored and the absence of mutations intended to create a solvent exposed glycan binding pocket, the simpler explanation for agonist activity is the unique conformation of the antibody rather than creation of a new glycan binding paratope.

While we feel that it is highly unlikely that the underlying mechanism of action for agonism by the iAbs described in this work involves glycan binding, the reviewer is correct that we cannot completely rule out this possibility. Therefore, we have added a statement addressing glycan binding to the discussion (see lines 439-443).

Second, if the dimerization benefit for the antibodies studied is simply improving affinity for the same epitope, then this should be clearly outlined in the text and supported by a main text figure. Figure S5 showing the comparisons of the i-shaped antibodies to the natural ligands for the receptors studied would be more appropriate as a main text figure.

Figures 2D and 2E of the new version of the main text figures show that engineering the Fab dimerization interface into the anti-OX40 panel does not affect antigen binding both by surface plasmon resonance (SPR) and cell surface binding. Thus, the data clearly indicate that improvement in affinity is not the underlying mechanism of action for receptor agonism. To address the second part of this comment, we have moved the native ligand comparison for OX40 from Figure S5 to the main text, now Figure 2C.

Third, in regard to the applicability of using these new agonists therapeutically, the authors did not indicate the number of individuals from whom primary cells were obtained. Instead, the number of replicate assays were shown. For example, on line 1155, there should be "N=X" next to individuals.

Both the figure legend and the methods section describing these experiments state that the primary cells were obtained from an individual donor. To make this further evident, we have added "(n=1)" next to this statement as requested by the reviewer.

Fourth, the introduction and discussion could benefit from revisions to improve clarity and impact for translation of these results to basic/translational researchers. As is, it is written for a targeted audience and may be dense for nature communications readership.

Fourth, the introduction and discussion could benefit from revisions to improve clarity and impact for translation of these results to basic/translational researchers. As is, it is written for a targeted audience and may be dense for nature communications readership.

On the point of audience, it is unclear from this comment exactly what targeted audience the Reviewer feels that the writing is skewed towards. In our view, the Introduction and Discussion sections, as well as the diversity of experiments in the Results section, span an array of fields that make this work of interest to the basic/translational scientific community. There are elements in this work that touch receptor biochemistry, cell biology, structural biology, biophysics, protein engineering, and, of course, drug discovery and development. The Introduction addresses a broad range of topics including the current state of therapeutic agonist clinical development, challenges associated with this class of drugs, beneficial properties of antibodies that have the potential to overcome these challenges, as well as an overview of approaches that have been made to improve antibody-based agonists (including the present study). The Discussion not only highlights some of our more impactful findings, discusses some of the pitfalls of this work, and suggests some important next steps, but also frames the work in the context of the current state of biotherapeutic agonists.

We believe that we have sufficiently and accurately presented this work in the broader context of translation, especially considering its early stage in the drug development process, as we appropriately acknowledge in the manuscript. This current comment 4 is similar to the Reviewer's previous comment 9 in the first review. In an effort to address that original comment in the first review we added an extensive paragraph to the Discussion. That added paragraph

articulates what we as drug developers view as key elements for translation (i.e. developability, PK, immunogenicity, and the like), as well as the potential impact of this approach and context for where it sits in the greater arch of biotherapeutic formats and receptor agonists. This current comment 4 indicates that, in the Reviewer's eyes, our added discussion of these topics was insufficient. Yet the vague nature of the comment makes it challenging for us to understand how our previously added paragraph doesn't meet the Reviewer's expectation, and accordingly how we could better address it.

Perhaps there is also a stylistic difference involved here, and the acknowledged denseness of the paper does make it some effort to follow. This work is complex, and while we've made our best efforts to present it in a clear manner, we appreciate how it may come across as overly dense. On that topic, we have made some minor edits to both the Introduction and Discussion to cut some superfluous words and phrases in hopes of making those sections slightly less wordy and more readable.